# Trypanosomes lack a canonical EJC but possess an UPF1 dependent NMD-like pathway

Bernardo Papini Gabiatti[1], Eden Ribeiro Freire[2], Johanna Odenwald[1], Janaina de Freitas Nascimento[2¤], Fabiola Holetz[3], Mark Carrington[2], Susanne Kramer[1]*, Martin Zoltner[4]*

1 Department of Cell and Developmental Biology, University of Würzburg, Würzburg, Germany, 2 Department of Biochemistry, Cambridge University, Cambridge, United Kingdom, 3 Carlos Chagas Institute (ICC), FIOCRUZ/PR, Curitiba, Brazil, 4 Department of Parasitology, Faculty of Science, Charles University in Prague, Biocev, Vestec, Czech Republic

¤ Current address: Parasitology Department, Instituto de Ciências Biomédicas II, Universidade de São Paulo, São Paulo, Brazil
* martin.zoltner@natur.cuni.cz (MZ); susanne.kramer@uni-wuerzburg.de (SK)

## Abstract

The exon junction complex (EJC) is a key player in metazoan mRNA quality control and is placed upstream of the exon-exon junction after splicing. Its inner core is composed of Magoh, Y14, eIF4AIII and BTZ and the outer core of proteins involved in mRNA splicing (CWC22), export (Yra1), translation (PYM) and nonsense mediated decay (NMD, UPF1/2/3). *Trypanosoma brucei* encodes only two genes with introns, but all mRNAs are processed by *trans*-splicing. The presence of three core EJC proteins and a potential BTZ homologue (Rbp25) in trypanosomes has been suggested to adapt of the EJC function to mark *trans*-spliced mRNAs. We analysed trypanosome EJC components and noticed major differences between eIF4AIII and Magoh/Y14: (i) whilst eIF4AIII is essential, knocking out both Magoh and Y14 elicits only a mild growth phenotype (ii) eIF4AIII localization is mostly nucleolar, while Magoh and Y14 are nucleolar and nucleoplasmic but excluded from the cytoplasm (iii) eIF4AIII associates with nucleolar proteins and the splicing factor CWC22, but not with Y14 or Magoh, while Magoh and Y14 associate with each other, but not with eIF4AIII, CWC22 or nucleolar proteins. Our data argue against the presence of a functional EJC in trypanosomes, but indicate that eIF4AIII adopted non-EJC related, essential functions, while Magoh and Y14 became redundant. Trypanosomes also possess homologues to the NMD proteins UPF1 and UPF2. Depletion of UPF1 causes only a minor reduction in growth and phylogenetic analyses show several independent losses of UPF1 and UPF2, as well as complete loss of UPF3 in the Kinetoplastida group, indicating that UPF1-dependent NMD is not essential. Regardless, we demonstrate that UPF1 depletion restores the mRNA levels of a PTC reporter. Altogether, we show that the almost intron-less trypanosomes are in the process of losing the canonical EJC/NMD pathways: Y14 and Magoh have become redundant and the still-functional UPF1-dependent NMD pathway is not essential.

**Data availability statement:** All proteomics data have been deposited at the ProteomeXchange Consortium via the PRIDE partner repository [1] with the dataset identifier PXD050946.

**Funding:** The project was funded by a bilateral GACR/DFG grant (project IDs.: 21-19503J and KR4017/9-1; to M. Z. and S. K., respectively) and the DFG grant KR4017/8-1 to S.K.

**Competing interests:** The authors have declared that no competing interests exist.

## Introduction

Trypanosomes are single celled parasites that split from other kinetoplastid lineages, like the free-living Bodonids, about 500 million years ago [2], while the order of Kinetoplastida itself branched off early after eukaryogenesis. As a consequence, trypanosomes evolved a range of unique biological features, including an unusual genome structure and gene expression pathway. Trypanosomes have a gene dense genome with almost no regulatory regions exerting transcriptional control and only two conserved intron-containing genes, DBP2B and PAP1, which have essential functions [3]. For the majority of genes, promoters are absent and RNA polymerase II transcription start sites are defined epigenetically [4]; the sole exception are the abundant mRNAs encoding the major surface coat proteins that are transcribed by RNA polymerase I [5]. All genes are transcribed as long polycistronic transcription units, subsequently processed by *trans*-splicing. The 39-nucleotide long miniexon sequence of the capped spliced leader RNA is *trans*-spliced to the 5' end of the transcript, in a splicing reaction that is coupled to the polyadenylation of the upstream transcript [6–9]. The mRNA cap is of the trypanosome-unique type 4 [10,11] and is decorated with the nuclear cap binding complex that consists of a homologue to Cbp20 and three trypanosome-unique subunits [12]. Trypanosomes lack obvious homologues to most proteins of the TREX and TREX2 (TRanscription-EXport) complex, which in opisthokonts link transcription and nuclear mRNA processing with mRNA nuclear export. However, an exception is the nuclear, essential DEAD box RNA helicase Sub2, which, despite of the absence of other TREX components, still appears to function in mRNA export, as suggested by nuclear mRNA accumulation upon its depletion [13]. More recently, three additional trypanosome TREX2 components, potential Sac3, Thp1 and Sus1 homologs, were suggested but remain functionally uncharacterized [14][15]. mRNA export is done via the canonical Mex67/Mtr2 export factor [16,17]. Unusually, Mex67 is associated with the small GTPase Ran, and trypanosomes have no homologue to the RNA helicase Dbp5 which usually releases the mRNA from the mRNP at the cytoplasmic site of the pore, suggesting that mRNA transport may be energetically coupled to the Ran gradient rather than to RNA helicase activity [18]. Moreover, mRNA export can be initiated co-transcriptionally [19] and a fraction of unspliced, dicistronic mRNAs reaches the cytoplasm [19–21], both arguing against a tight quality control system at the level of mRNA export.

Given this divergence in nuclear mRNA metabolism, it comes as a surprise that trypanosomes possess conserved homologues to proteins of the exon junction complex (EJC) and of the nonsense mediated decay (NMD) pathways. In opisthokonts, both the EJC and the NMD pathway fulfill important quality control functions for newly transcribed mRNAs. The EJC consists of three core subunits: the DExH/D-box RNA helicase eIF4AIII (EIF4A3/ Fal1/ DDX48), Y14 (RBM8A) and Magoh; a fourth subunit BTZ (Barentsz/MNL51/CASC3), suggested as peripheral EJC factor [22,23] is metazoan specific. The EJC assembles early during splicing. eIF4AIII, which has the main RNA binding activity of the EJC, is recruited to the mRNA by the essential spliceosomal factor CWC22 [24] and this binding is stabilized by the Y14/Magoh heterodimer which inhibits the ATPase activity of eIF4AIII, thus locking it to the transcript [25,26]. Y14 has an RNA recognition motif (RRM) containing domain which engages in an interaction with Magoh rather than with RNA [27,28]. BTZ, the fourth peripheral EJC subunit, unique to metazoans, joins after the release of the spliceosome [29]. The EJC positively affects mRNA processing and export: all EJC core proteins provide binding platforms for a range of peripheral factors involved in the regulation of gene expression [30]; Magoh, for example, interacts with the TREX complex [31]. In metazoans, the EJC accompanies the mRNA to the cytoplasm, where it is disassembled during the pioneer round of translation by the ribosome associated factor PYM [32]. Its cytoplasmic function is in the

identification of premature stop codons (PTCs): stop codons > 50 nucleotides upstream of the last exon-exon junction are considered PTCs and prime the mRNA for degradation by the NMD pathway [33–36]. This is mediated by the NMD factor UPF3b (Up-frameshift suppressor protein 3b) which directly interacts with the EJC core subunits and recruits further NMD factors, including the core factors UPF1 and UPF2 [37,38]. While the EJC-dependent NMD pathway appears to be specific to some metazoans, EJC-independent NMD is largely conserved across eukaryotes, employing the same core factors UPF1, UPF2 and UPF3. Instead of the splice-site position, EJC-independent NMD analyses a range of transcript features to distinguish a PTC from a physiological stop codon, as for example the distance to the poly(A) tail [39]. The eukaryotic release factors eRF1 and eRF3, which are recruited to the A-site of the ribosome at a stop codon, can either bind to Poly(A) binding protein or to NMD factors. While closeness of the stop codon to the poly(A) tail favors binding to Poly(A) binding protein, longer distances are indicative of a PTC and thus promote recruitment of NMD factors. The mechanisms of EJC-independent NMD are not entirely understood [40,41].

Trypanosomes possess homologues to the EJC proteins Magoh, Y14 and eIF4AIII [42,43]. For Y14, the homology is restricted to the RRM domain and both Magoh and Y14 lack amino acids required for eIF4AIII interaction [42], while eIF4AIII catalytic residues are conserved in trypanosomes (S1A Fig). There is experimental evidence for an interaction between these subunits in two trypanosomatid species, *T. brucei* (Tb) and *T. cruzi* (Tc): In yeast-two-hybrid assays, TcY14 interacts with TcMagoh and even with human Magoh [42]. In two independent pulldown studies, TbY14 captured TbMagoh but not TbeIF4AIII [42] and TcMagoh captured TceIF4AIII and *vice versa* [44]. Moreover, a protein with a nuclear transport factor (NTF2) domain (NTF2-like=NTF2L) was captured by both Y14 [42] and eIF4AIII [44], suggesting a connection of the EJC with nuclear transport. A recent study has analysed the interactomes from several Trypanosomatid candidates from affinity purification experiments and describes a complex series of associations connecting splicing, export and translation [44], including connections between the EJC and Sub2, the EJC and NTF2L and Sub2/NTF2L with Mex67/Mex67b [44]. This same study identified a putative trypanosomatid orthologue of BTZ/MNL51 (annotated as Rbp25 in *T. brucei*), which is exclusively captured by Sub2 [44]. RNAi depletion of eIF4AIII is lethal [43], while RNAi depletion of Magoh and Y14 has only a minor growth phenotype [42]. Taken together, the data support the presence of a functional EJC in trypanosomes and have fueled speculations that the EJC may constitute the missing regulator of mRNA export in trypanosomes [42–46]. With a possible role of marking *trans*-spliced mRNAs, it would link splicing and nuclear export pathways and promote the export of mature mRNAs.

Trypanosomes also have orthologues to the NMD proteins UFP1 and UPF2, but whether the parasites have a functional NMD pathway remains controversial. UPF1, that carries conserved helicase catalytic residues (S1B Fig), coprecipitates with UPF2 in an RNA-independent manner, and also with the poly(A) binding protein, in an RNA-dependent manner, indicative of a conserved function [47]. Moreover, the presence of a PTC in a reporter mRNA decreased mRNA stability, consistent with the presence of an NMD pathway [47]. Likewise, when the endogenous gene encoding VSG (the major, highly abundant variant surface glycoprotein) was expressed from an ectopic site, with PTCs ranging from early to late, mRNA levels were clearly reduced in dependency of PTC position, with a maximum reduction of 95% in the *VSG* mRNA with the earliest PTC [48]. However, the UPF1 dependence of this effect remains unclear and, in fact, RNAi depletion of neither UPF1 nor UPF2 caused any growth effects [47]. An argument against the presence of an NMD pathway comes from a recent sequencing study with allele specific resolution [49]: the authors looked at a range of transcripts that had endogenous, allele specific PTCs encoded in the DNA, but observed no difference in mRNA

abundance to the equivalent transcript from the non-mutated allele. In fact, the mRNAs with the PTC even had equal ribosome occupancy of the region preceding the PTC, ruling out any differences in translational initiation as well.

Apart from these reports in trypanosomes, there are few studies on NMD in protozoa. *Paramecium tetraurelia*, which has a genome that is rich in tiny introns, has all three UPF variants and for UPF1 and UPF3 a function in NMD was experimentally shown [50,51]. NMD is likely EJC-independent as the amino acids of UPF3 that are required for interaction with the EJC are not conserved [51]. Likewise, the intron-rich ciliate *Tetrahymena thermophila* does EJC-independent NMD with the aid of UPF1/2/3 [52]. *Giardia lamblia*, in contrast, is intron-poor and has only UPF1. There is experimental evidence for an UPF1 dependent NMD mechanism in *Giardia*, albeit not a very efficient one [53]. Interestingly, UPF1 upregulation decreased levels of selected mRNAs, including mRNAs encoding for life-cycle specific proteins, indicating an PTC-independent function of UPF1 in gene regulation [54].

Here, we set out to reinvestigate the trypanosome EJC and NMD components, employing a TurboID proximity labelling approach combined with a range of novel tools. In particular, we employ the auxin-inducible degron system (AID2) for fast protein depletion and antibody-independent protein localization by streptavidin imaging to avoid artefacts by antibody-accessibility problems [55]. Our data strongly indicate that the EJC has no function in the nuclear export of *trans*-spliced mRNAs. In fact, we observe a loss of a functional EJC complex in trypanosomes, and show that only eIF4AIII has essential, but EJC-independent functions. Furthermore, we show that UPF1-dependent NMD is functional in trypanosomes but not essential for cultured cells, as there were several independent losses of the NMD components UPF1-3 in the Kinetoplastea lineage, and UPF1 depletion caused only a minor growth effect.

## Materials and methods

### Cell culture

Trypanosoma brucei Lister 427 procyclic cells in logarithmic growth were used for all experiments. Cells were grown in SDM-79 supplemented with 5% FCS and 75 μg/ml hemin at 27°C and 5% $CO_2$. [56] Transgenic trypanosomes were kept under the respective selection (G418 disulfate (15 μg/ml), blasticidin S (10 μg/ml), puromycin dihydrochloride (1 μg/ml) and hygromycin B (25 μg/ml); for selection after a transfection, antibiotic concentrations were doubled). Growth was monitored by sub-culturing cells repeatedly to $10^6$ cells/ml and measuring densities 24 hours later using a Coulter Counter Z2 particle counter (Beckman Coulter). Transgenic trypanosomes were generated by standard procedures, relying on the transfection of linearized DNA by electroporation [57]; correct integration of plasmids/PCR products and/or correct protein expression was confirmed via PCR from genomic DNA and/or western blot, respectively.

### Auxin inducible protein degradation

The auxin-inducible degron system (AID2) [58] was adapted to trypanosomes and kindly provided by Prof. Mark Carrington, University of Cambridge (manuscript in preparation). Briefly, the mother cell line was made by integrating five proteins from *Oryza sativa, each* fused to a Ty1 epitope tag for western blot detection, into the tubulin locus of procyclic trypanosomes for endogenous expression, via two plasmid transfections. The first plasmid transferred the genes for *Os*SKP1, *Os*CUL1, *Os*RBX1; the second the genes for *Os*TIR1(F74G) and *Os*ARF. In this cell line, both alleles of the target protein were modified to express 3HA-AID2 fusion variants (either N or C terminal tagged), by two subsequent transfections using PCR tagging with the (modified) pPOTv7 system [59] with two different selection markers (one for each allele). Protein degradation was induced by adding 50 μM of 5-Ph-IAA (MedChem

Express, HY-134653) directly to the culture, from a 50 mM 5-Ph-IAA stock (in DMSO, stored at -20ºC for short-term use or -80ºC for long-term).

## Plasmids and PCR products

Detailed information on plasmids, PCR products and primers used in this study is available in S1 Table. To generate the knockouts, PCR products were created by amplification of the resistance marker with 80 nucleotides overhangs of the 5' and 3' untranslated region (UTR) of the gene of interest [60]. The expression of fusion proteins from the endogenous locus was done using the pPOTv7 system [59]. A cell line used as a negative control of the TurboID experiment expressed a codon-optimized eYFP-TurboID-HA [61] with the La protein NLS [62] integrated into the tubulin locus.

## Western blotting

Western blotting was performed using standard methods, usually loading about $5x10^6$ cells per lane. Primary antibodies were rat anti-HA (3F10, Roche) (1:1000) and anti-BiP (1:100,000) (kind gift from Jay Bangs) and secondary antibodies were IRDye® 680 RD and 800 CW rat and rabbit anti-goat (LI-COR) (1:30,000). Detection and quantification was done using the Odyssey Infrared Imaging System (LI-COR Biosciences, Lincoln, NE). For quantification, the average of a 3-pixel width line at the top and bottom of each band was subtracted from each pixel for background correction.

## Immunofluorescence

Imaging of TurboID-HA tagged proteins was done with anti-HA and fluorescent streptavidin, essentially as described [55]. Briefly, about $1 \times 10^7$ procyclic-form *T. brucei* cells, harvested at a density of $5 \times 10^6$ cells/ml, were washed once in 1 ml hemin- and serum-free SDM79, then fixed in 1 ml of 4% paraformaldehyde. After addition of 7 ml PBS supplemented with 20 mM glycine, cells were pelleted, resuspended in 150 µl PBS, and spread on poly-lysine-coated slides. Cells were permeabilized with 0.5% Triton X-100 in PBS, then rinsed in PBS, and blocked in 3% bovine serum albumin (BSA) in PBS for 30 min, followed by 60 min incubation with rabbit mAb-anti-HA C29F4 (1:500 dilution; Cell Signaling Technology) and Streptavidin-Cy3 (1:200 dilution; Jackson Laboratories) in PBS/3% BSA. Slides were washed in PBS (three times for 5 min), then incubated with anti-rabbit Alexa Fluor Plus 488 (1:500 dilution, A32731 Invitrogen) in PBS/3% BSA for 50 min and a further 10 min upon addition of 4',6-diamidino- 2-phenylindole (DAPI) (0.1 µg/ml). Slides were washed 3 × 5 min in PBS and embedded into ProLong Diamond Antifade Mountant (Thermo Fisher Scientific).

## Affinity purification *of* biotinylated proteins, mass spectrometry and analysis

Affinity purification of biotinylated proteins and tryptic digest and peptide preparation were performed as described [61], except that 1 mM biotin was added to the on-bead tryptic digests, to improve the elution of biotinylated peptides. Eluted peptides were analyzed by liquid chromatography coupled to tandem mass spectrometry (LC-MSMS) on an Ultimate3000 nano rapid separation LC system (Dionex) coupled to an Orbitrap Fusion mass spectrometer (Thermo Fisher Scientific).

Spectra were processed using the intensity-based label-free quantification (LFQ) in Max-Quant version 2.1.3.0 [63,64], searching the *T. brucei brucei* 927 annotated protein database (release 64) from TriTrypDB [65]. Analysis was done using Perseus [66] essentially as described in [67]. Briefly, known contaminants, reverse and hits only identified by site were filtered out.

LFQ intensities were $\log_2$-transformed and missing values imputed from a normal distribution of intensities around the detection limit of the mass spectrometer. A Student's $t$-test was used to compare the LFQ intensity values between the duplicate samples of the bait with an untagged control (WT parental cells) triplicate samples. The $-\log_{10}$ $p$-values were plotted versus the $t$-test difference to generate multiple volcano plots (Hawaii plots). Potential interactors were classified according to their position in the Hawaii plot, applying cut-off curves for significant class A (SigA; FDR = 0.01, s0 = 0.1) and significant class B (SigB; FDR = 0.05, s0 = 0.1). The cut-off is based on the false discovery rate (FDR) and the artificial factor s0, which controls the relative importance of the $t$-test $p$-value and difference between means (at s0 = 0 only the $p$-value matters, while at non-zero s0 the difference of means contributes). Perseus was also used for principal component analysis (PCA), the profile plots and to determine proteins with similar distribution in the plot profile using Pearson's correlation. Gene Ontology analysis was done on TriTrypDB [65], with the option to use only slim terms and the $p$-value was adjusted with a Bonferroni correction. All proteomics data have been deposited at the ProteomeXchange Consortium via the PRIDE partner repository [1] with the dataset identifier PXD050946.

### Single molecule fluorescence in situ hybridization (smFISH)

smFISH was done as previously described [68]. The sequences used to design the Affymetrix FISH probes are in S1 Table.

### Microscopy and quantification of microscopy data

For Fig 2B, S2B Fig and S6 Fig, images were acquired using a DMI8 widefield microscope (Thunder Imager, Leica Microsystems, Germany) with a HCX PL APO CS objective (100x, NA = 1.4, Leica Microsystems) and Type F Immersion Oil (refractive index = 1.518, Leica Microsystems). The microscope was controlled using LAS-X software (Leica Microsystems). Samples were illuminated with an LED8 light source (Leica). Excitation light was selected by using the following filter sets: DAPI - Ex 391/32 nm; DC 415nm; Em 435/30 nm (used for DAPI). FITC - Ex 436/28 nm; DC 500 nm; Em 519/25 nm. mCherry – Ex 578/24 nm; DC 598 nm; Em 641/78 nm. Cy5 – Ex 638/31 nm; DC 660 nm; Em 695/58 nm. Images were captured using a K5 sCMOS camera (Leica, 6.5 μm pixel size). Z-stacks (75 slices, 140 nm step size) were recorded. For Fig 2A and S5 Fig, a custom-built TILL Photonics iMIC microscope equipped with a 100x, 1.4 numerical aperture objective (Olympus, Tokyo, Japan) and a sensicam qe CCD camera (PCO, Kehlheim, Germany) was used. eYFP was monitored with the FRET-CFP/YFP-B-000 filter, mCherry with the ET-mCherry-Texas-Red filter and DNA with the DAPI filter (Chroma Technology CORP, Bellows Falls, VT). Z-stacks (75 slices, 140 nm step size) were recorded. All images were deconvolved using Huygens Essential software (SVI, Hilversum, The Netherlands). All data were analysed with Fiji [69].

### BLASTp screen for presence of UPF sequences

The UPF1 and UPF2 sequences from Trypanosomatids and other kinetoplastids were obtained by using *T. brucei* (Tb927.5.2140 = UPF1; Tb927.11.6410 = UPF2) protein sequence as query on BLASTp [70] searches on TritrypDB [65], GenBank [71] and ENA [72] databases. The absence of UPF3 in Kinetoplastida was confirmed by blasting with human UPF3. The UPF sequences from other eukaryotes were obtained searching the same databases with human UPF sequences. Sequences from organisms that were not available on the online searches were obtained by downloading full transcriptome data and using it on local BLASTp searches on BioEdit 7.2.5 [73]. Sequence results with e-value less than $1\times10^{-3}$ were discarded. All sequences are available in S3 Table. Alignments were visualized using Jalview [74].

### Flow cytometry and Northern blot analysis of PTC reporters

Northern blots were done as described [75]. For flow cytometry, data were acquired using FACScan (Becton Dickinson, Franklin Lakes, NJ) and analysed using Cell Quest V3.3 software. Flow Check Fluorospheres (Beckman Coulter, Pasadena, CA) were used for normalisation.

### RT-qPCR analysis of PTC reporters

eGFP sequences, either with a native stop codon or premature stop codons at positions 60 and 120, were integrated into tubulin locus for constitutive expression. All cell lines were confirmed by Sanger sequencing. UPF1 degradation was induced for 2 hours with $50\,\mu M$ 5-Ph-IAA and RNA was extracted (RNeasy Mini Kit, Qiagen, according to manufacturer's protocol and including the optional in column DNAse digestion step). RNAs were diluted to $50\,ng/\mu l$ and used for RT-qPCR with iTaq Universal One-Step RT-qPCR Kit (BIORAD) in a StepOne™ Real-Time PCR System (ThermoFisher). The reaction was done in $10\,\mu l$ total volume using $100\,ng$ RNA, $1\,\mu M$ of each forward and reverse oligo and the cycler program of $50\,°C$ for $20\,min$, $95\,°C$ for $1\,min$ and 40 cycles of $95\,°C$ for $15\,sec$ and of $60\,°C$ for $10\,sec$. Each sample was analysed in triplicates. GAPDH oligos were taken from [76]. eGFP oligos were designed using the IDT PrimerQuest™ tool (sequences in S1 Table). The CT values for eGFP were controlled for loading with GAPDH ($\Delta CT$) and the average value of the native stop codon without 5-Ph-IAA treatment was set as parental for the calculation of the 2-$\Delta\Delta CT$.

## Results and discussion

### eIF4AIII is essential, Magoh and Y14 are not

We attempted to generate genomic knock-out cell lines for Y14, Magoh and eIF4AIII. We transfected PCR products consisting of a resistance marker flanked by 80 nucleotides with homology to the respective 5'- and 3' regions to replace both alleles of each gene, in two subsequent transfections [59]. For Magoh and Y14, multiple double-resistant cell lines were readily obtained, and the absence of the wild type gene was confirmed by PCR analyses (S2A–S2C Fig). Furthermore, we could delete both Magoh alleles in a Y14 knockout strain, creating a double knockout. All knockout strains displayed only a minor growth phenotype (10% reduced growth rate for the Y14 knockout, 20% for the Magoh knockout and Y14/Magoh double knockout) (Fig 1A) which could not be assigned to a block in a specific cell cycle phase (S2D Fig). Silencing of Magoh in *Paramecium* or knockout of Y14 in *Y. lipolytica* caused changes in intron retention, indicating a function in splicing [77,78]. We therefore analysed the *T. brucei* Y14/Magoh double knockout for splicing defects, by probing for exon and intron of the intron-containing mRNA *DBP2B* of *T. brucei*, with dual colour single molecule smFISH [68]. We observed no differences in numbers or cytoplasmic/nuclear RNA distribution between wild type cells and Y14/Magoh double knockout strains, indicating that Y14 and Magoh are not required for processing, export or stability of *cis*-spliced mRNA (S3 Fig). Surprisingly, we were able to detect both intron-containing mRNAs and intron alone in the cytoplasm, even in wild type cells, representing additional evidence for the absence of a tight quality control system of mRNA nuclear export in trypanosomes.

For eIF4AIII, the first allele could readily be deleted, but multiple trials to knock out the second allele failed, indicating that eIF4AIII is essential. We therefore employed the auxin-inducible degron system (AID2) [58] which was recently adapted to trypanosomes (Mark Carrington, manuscript in preparation). Briefly, both eIF4AIII alleles were replaced by eIF4AIII fused to *Os*AID2-3HA in a cell line constitutively expressing the five *Oryza sativa* proteins

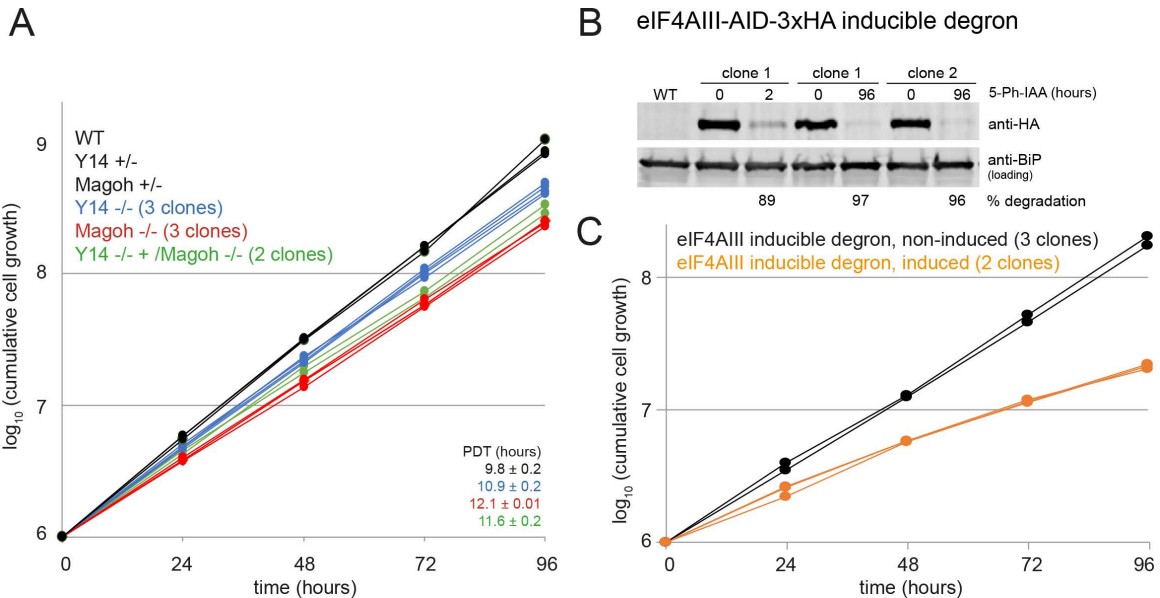

**Fig 1. Growth of Y14/Magoh knockout strains and cells upon auxin-induced degron depletion of eIF4AIII. A)** Growth of wild type cells (WT), heterozygote (+/-) and homozygote (-/-) Y14 and Magoh knockout cells and Magoh/Y14 double knockout cells was monitored over 96 hours. The population doubling time (PDT) is indicated on the right (mean of PDT of all clones ± standard deviation) **B and C)** Both eIF4AIII alleles were replaced by eIF4AIII-AID2-3HA in a cell line expressing all proteins required for the AID2 degron system. eIF4AIII degradation was induced by the addition of 50 μM 5-Ph-IAA. The loss in eIF4AIII-AID2-3HA protein was monitored on a western blot, loaded with protein samples taken after 2 hours of induction, and protein samples of three different clonal cell lines taken after 96 hours of induction (B). For two clonal cell lines, the growth was monitored in the presence and absence of 5-Ph-IAA (C). Note that the degron mother cell line has a slightly reduced growth rate.

required for the degron system (details in material and methods). The absence of a wild type eIF4AIII allele was confirmed by PCR (S4A–S4C Fig). Induction of the system with the auxin analogue 5-Ph-IAA resulted in almost complete degradation of eIF4AIII within 2 hours which was not reversed (Fig 1B, Fig 1 in S1 Raw Images). Cell growth was significantly reduced (Fig 1C). No specific cell cycle phase was affected, albeit there was a small increase in the number of "zoids", anucleate cells that have a kinetoplast only (S5 Fig).

In conclusion, eIF4AIII is essential in cultured trypanosomes, but Magoh and Y14 are not, arguing against an exclusive role of these proteins within the same complex. This is in agreement with data from previous RNAi experiments [42,43].

## eIF4AIII, Magoh and Y14 are nuclear proteins, with eIF4AIII being preferentially nucleolar

To determine the subcellular localization of Magoh, Y14 and eIF4AIII, we expressed these proteins as eYFP fusions, by modifying one endogenous allele [59]. Magoh and eIF4AIII were expressed as C-terminal fusions, Y14 was tagged at the N-terminus. The RNA helicase mChFP-DHH1, expressed from the endogenous locus, served as a cytoplasmic control. eYFP fusions of all three proteins were highly enriched in the nucleus, while DHH1 was entirely cytoplasmic (Fig 2A and S6 Fig). Magoh, Y14 and eIF4AIII eYFP fusion proteins were present in both the nucleolus (defined by the absence of DAPI stain) and the nucleoplasm. However, we noticed two differences between Y14/Magoh and eIF4AIII localisations: (i) eIF4AIII was more enriched in the nucleolus, while Y14 and Magoh were more evenly spread throughout the entire nucleus and (ii) a small fraction of eIF4AIII, but not of Magoh or Y14, was

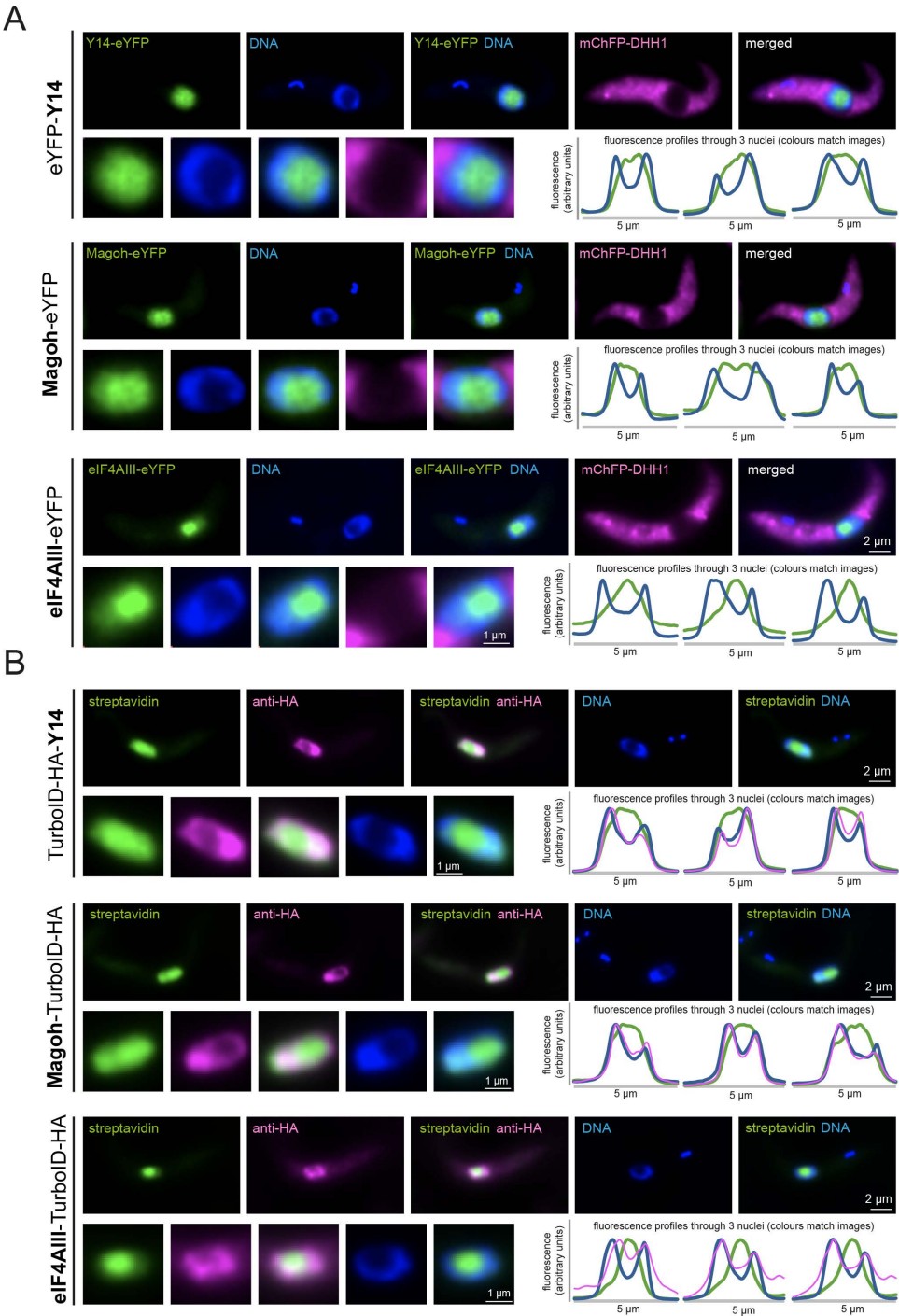

**Fig 2. Localisation of the EJC complex components. A)** Cells expressing eYFP fusion proteins of Y14, Magoh and eIF4AIII in a cell line co-expressing mChFP-DHH1 were fixed and imaged; DNA was stained with DAPI. Z-stack projections (75 slices a 140 nm, sum slices) of one representative cell are shown; the corresponding nucleus is shown enlarged. For each cell line, fluorescence profiles through three nuclei are shown. More images, including images of cells treated with actinomycin D, sinefungin or starvation are in S5 Fig **B)** Cells expressing eIF4AIII-HA-TurboID were labelled with streptavidin-Alexa 488 (green) and anti-HA (secondary antibody Alexa 594 labeled, shown in pink). DNA was stained with DAPI (blue). Deconvolved Z-stack projections (100 slices a 100 nm) of one representative cell are shown; the corresponding nucleus is shown enlarged. For each cell line, fluorescence profiles through three nuclei are shown.

cytoplasmic (see fluorescence-profiles across the nuclei in Fig 2A), in agreement with eIF4AIII shuttling between nucleus and cytoplasm, as previously reported [79]. The data agree with localizations obtained by the genome-wide localization project TrypTag [80] and for Y14 also with immunofluorescence data using fluorescent anti-IgG against Y14 fused to the TAP tag [42].

We observed no major changes in protein localization when we manipulated mRNA metabolism with actinomycin D (inhibition of transcription), sinefungin (inhibition of *trans*-splicing) or PBS-starvation (global inhibition of translation) (S6 Fig). All three proteins remained highly enriched in the nucleus. We observed a minor increase in cytoplasmic localization for eIF4AIII at starvation stress; as this protein appears to shuttle the inhibition of global mRNA metabolism may slightly shift the distribution of eIF4AIII between the two compartments (S6C Fig). Moreover, Y14 fluorescence appeared shifted from the nucleoplasm to the nucleolus at sinefungin treatment (S6A Fig).

For eIF4AIII (alias Hel46), our data contradict a range of immunofluorescence experiments done in *T. cruzi* and *T. brucei*, with antibodies against the native protein or anti-Protein A to a PTP-tagged version [79]. The previous authors reported a dominant cytoplasmic localization of eIF4AIII, with a nuclear localization being enforced only when nuclear export was blocked or a putative NES removed. It is possible, that the nuclear localization we and others observe with fusion proteins to GFP derivatives is an artefact of the tag, but, we consider this unlikely, because eYFP tagging at both the C- and the N-terminus results in the same nuclear localization [80] and, we do know that a C-terminal fusion to AID2-3xHA (which is similar in size to eYFP) is fully functional in the absence of wild type eIF4AIII (Fig 1B and 1C, Fig 1 in S1 Raw Images). An alternative explanation for this discrepancy is that antibodies fail to label eIF4AIII inside the nucleolus, because they cannot penetrate phase-separated areas. We have previously observed this for the *T. brucei* nucleolar protein NOG1 and for a range of proteins localized to other phase-separated compartments [55]. In fact, an older study shows marked differences in eIF4AIII localization between antibody detection (nuclear) or detection of an eYFP-fusion (nucleolus) [43]. We have recently proposed the use of biotin proximity labeling in combination with streptavidin to overcome antibody-accessibility problems: the target protein is fused to the biotin ligase TurboID and the resulting (auto-)biotinylation of this bait and adjacent proteins is detected by fluorescent streptavidin, which readily enters phase-separated areas [55]. Hence, we endogenously expressed TbeIF4AIII C-terminally fused to a tandem tag of the biotin-ligase TurboID [81] and 3HA epitope tags. We imaged eIF4AIII localization in parallel with both fluorescent streptavidin and anti-HA (Fig 2B and S7 Fig). The streptavidin signal was in the nucleolus and partially in the nucleoplasm, resembling the eYFP signal. In contrast, anti-HA weakly stained the nucleoplasm and partially the cytoplasm but was excluded from the nucleolus, proving that eIF4AIII can indeed not be correctly localized with immunofluorescence detection and providing an explanation for the discrepancies between the data. We have repeated the experiment with Magoh and Y14 (Fig 2B and S7 Fig). With streptavidin, we obtained both nucleolar and nucleoplasmic localisation, in agreement with the eYFP fluorescence data (Fig 2A), while the anti-HA signal was restricted to the nucleoplasm and absent from the nucleolus (Fig 2B). Interestingly, the cytoplasmic HA signal was significantly stronger for eIF4AIII than for Magoh and Y14 (images and profiles in Fig 2B), providing further evidence for eIF4AIII being a shuttling protein. This was not observed for streptavidin; a possible reason is that streptavidin detects all eIF4AIII-proximal proteins and the majority of eIF4AIII interactors may not be shuttling.

To summarise, imaging with either eYFP or streptavidin localized Magoh, Y14 and eIF4AIII to the nucleoplasm and the nucleolus, with eIF4AIII being mostly nucleolar and Y14 and Magoh being more equally spread between the nucleolus and the nucleoplasm. eIF4AIII,

but not Y14 and Magoh, was also partially cytoplasmic. As the exon junction complex functions in RNA splicing and processing, the expected localization is nucleoplasmic. The observed nucleolar localization of eIF4AIII, and to a lesser extent of Magoh and Y14, therefore contradicts such canonical roles. Moreover, the differences in localizations argue against the exclusive presence of the three proteins in one complex.

## eIF4AIII and Magoh/Y14 associate with different cohorts of proteins

To explore the interactome of eIF4AIII, Y14 and Magoh, we employed the available cell lines that expressed the TurboID-HA fusion proteins (Fig 2B). For each bait protein, we carried out two independent streptavidin-affinity purifications and subjected peptides eluted by on-bead tryptic digest to liquid chromatography coupled to tandem mass spectrometry (LC-MSMS) analyses. Intensities obtained from label-free quantification formed distinct replicate clusters in a principal component analysis for all bait and control groups (Fig 3A). For Magoh and Y14, 97 and 367 protein groups (Fig 3B and 3C; S2 Table), respectively, were found enriched (SigB; see methods section) indicating a considerable degree of bystander-labelling. However, Magoh was enriched in the Y14 TurboID and vice versa, strongly suggesting interactions between these two factors. eIF4AIII was absent from the Y14 and Magoh interactomes and, consistently, neither Magoh nor Y14 were among the 131 protein groups enriched (SigB) in the eIF4AIII TurboID experiment (Fig 3B and 3C; S2 Table). Instead, two nucleolar proteins, CWC22 and NOM1 (Sgd1p in yeast), were specifically enriched with eIF4AIII, but not with Y14 or Magoh. Both are homologs of known eIF4AIII interactors in opisthokonts [24,82]. Interestingly, CWC22 is a spliceosomal factor, required for EJC formation in mammalian cells [24] and NOM1 was proposed to function, EJC independently, in pre-ribosomal RNA processing [82]. The potential BTZ ortholog, RBP25, was enriched for all three bait proteins. Further candidate interactors are listed in S2 Table but remain at a low confidence level due to apparent high-level non-specific background, which may be partly attributed to the high abundance of the three bait proteins [83]. GO-term enrichment analyses for the three bait proteins (Fig 3D; S2 Table) reveals differences between eIF4AIII and Y14/Magoh. Whilst all baits show similar enrichment of the GO component 'nucleus', components 'nucleoplasm' and 'nucleosome' are enriched exclusively for Y14 and Magoh, while 'nucleolus' is unique for eIF4AIII. However, it is unclear whether these differences reflect different interactions or are merely reflecting bystander labeling profiles evoked by the more dominant eIF4AIII nucleolar localization.

Of interest was the absence of the NTF2L protein from the TurboID interactomes of all three EJC proteins, even though the protein was previously coprecipitated with *T. brucei* Y14 [42] and *T. cruzi* eIF4AIII [44]. Given that all three EJC complex proteins have exclusive (Magoh and Y14) or dominant (eIF4AIII) nuclear localization, while NTF2L is in the cytoplasm [80], it is possible that the interactions observed in the immunoprecipitations are an *in vitro* artefact of the cell lysis that allows interactions of proteins that are normally separated. However, this does not fully rule out an artefactual NTF2L absence in the TurboID. RNAi depletion of NTF2L caused only a minor growth effect and no nuclear mRNA accumulation, arguing against any essential function [44]. We show that even a fast reduction of NTF2L protein to 10% using the AID2 system caused only a minor reduction in growth (S4D Fig and S8 Fig, Fig 3 in S1 Raw Images). Our data provide no evidence for an essential function of NTF2L, within the EJC or elsewhere, in cultured procyclic trypanosomes.

## UPF1 dependent NMD is functional but not essential

The absence of a canonical EJC does not exclude the presence of an NMD pathway: NMD is EJC-independent in the majority of eukaryotes. *T. brucei* has homologues to UPF1 and

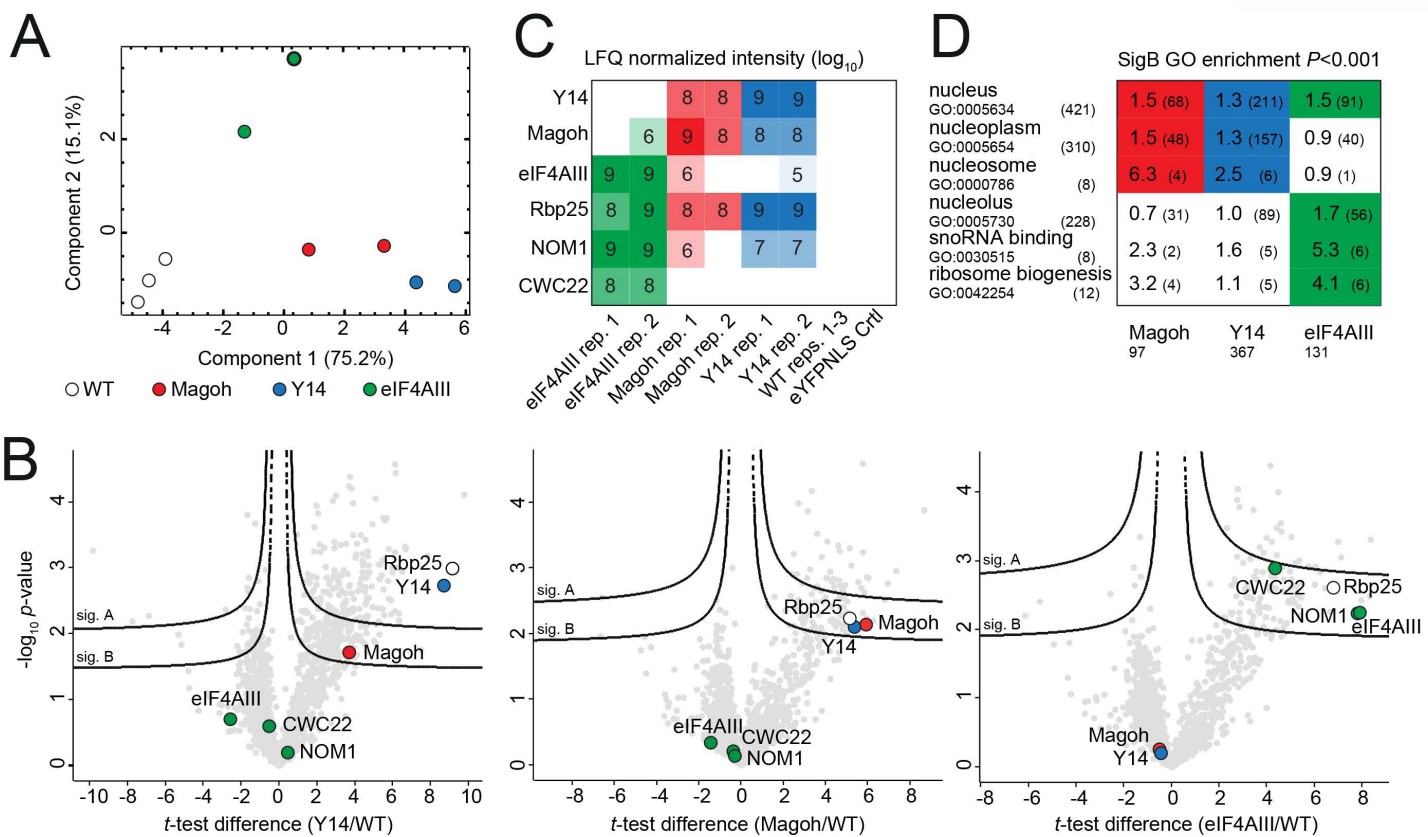

**Fig 3. Magoh, Y14, and eIF4AIII TurboID interactomes.** **A)** Principal component analysis (PCA) of the LFQ values obtained from TurboID duplicate experiments with Magoh (red), Y14 (blue), eIF4AIII (green) and three replicate parental (WT) controls (white). **B)** Corresponding Hawaii-plot (multiple comparative volcano plots) for statistical analysis of TurboID experiments. To generate the volcano plots, the $-\log_{10}P$-value was plotted versus the $t$-test difference, comparing each respective bait experiment to the parental control. Potential interactors were classified according to their position in the plot, applying two statistical cut-off curves, termed SigA and SigB (see Methods section). Selected protein groups are coloured (green for eIF4AIII and previously described interactors; red for Magoh; blue for Y14; white for the BTZ homolog RBP25). **C)** Diagram showing $\log_{10}$-transformed normalized LFQ intensities of baits and high-confidence interactors for each experiment replicate (color coded as in (A); blank cells indicate no detection) including a nuclear eYFP-TurboID control (eYFP_NLS; see methods section). **D)** Diagram showing selected GO terms enriched within the SigB cut-off from any of the three bait experiments (color coded as in (A)) with $p$-values < 0.001. Numbers in each cell indicate the enrichment factor and the intersection size (number of protein groups with the respective GO-term within SigB) in brackets. The selection size (total number of protein groups in SigB) is indicated at the bottom; the category size (total number of GO-term occurrence among all 927 detected protein groups) is indicated on the left next to GO IDs; white cells indicate no or insignificant GO enrichment. For all proteomics data and complete GO enrichment analyses see S2 Table.

UPF2 that interact, but RNAi depletion in either procyclic or bloodstream form trypanosomes had no effect on growth or the transcriptome [47]. Consistently, we observed only a minor growth defect upon reducing UPF1 levels to approximately 2% via the AID2 degron (Fig 4A and 4B, Fig 2 in S1 Raw Images and S4E Fig). We specifically looked for changes in intron retention in the UPF1 depleted cells with smFISH, by probing for *DBP2B*, as intron retained *DBP2B* contains a stop codon downstream of the intron but detected no differences (S3D Fig). Next, we expressed an eGFP reporter gene with either an early PTC (position 60), a late PTC (position 120), or the complete ORF (native, stop at position 240) constitutively from the tubulin locus (Fig 4C). While both premature stop codons resulted in the expected loss of GFP protein (Fig 4D), northern blotting revealed that the presence of a PTC significantly reduced the reporter mRNA levels in a position specific manner (Fig 4E, Fig 4 in S1 Raw Images): Early and late PTC evoked approximate mRNA decreases of 94% and 37%, respectively, when compared to the mRNA encoding native eGFP. This is consistent with

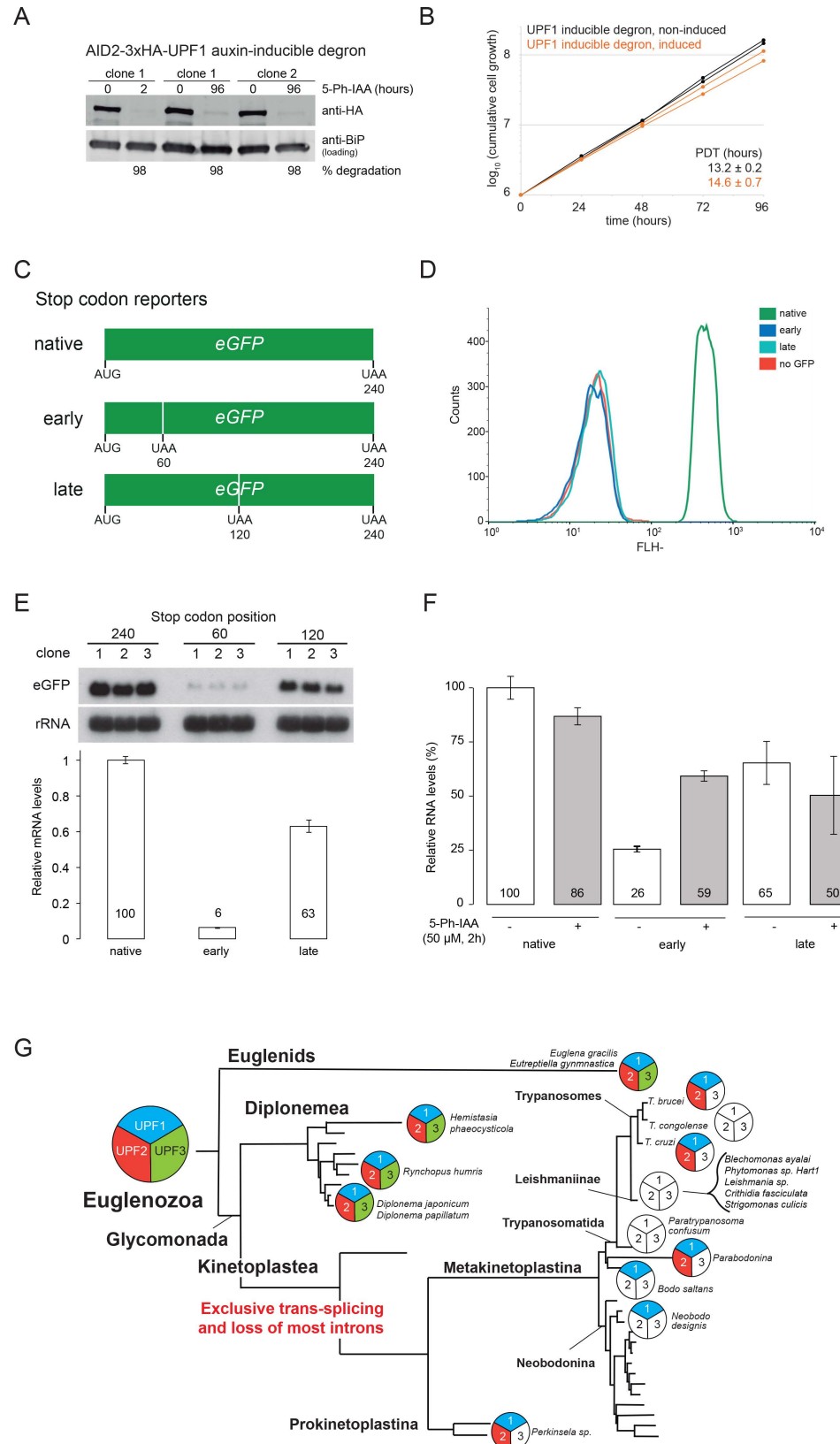

**Fig 4. NMD in Kinetoplastea. A and B)** Both UPF1 alleles were replaced by UPF1-AID2-3HA in a cell line expressing all proteins required for the AID2 degron system. UPF1 degradation was induced by the addition of 50 µM 5-Ph-IAA. The loss in UPF1-AID2-3HA protein was monitored on a western blot and quantified (A). For two clonal

cell lines, growth was monitored in the presence and absence of 5-Ph-IAA (B). Note that the degron mother cell line has a slightly reduced growth rate. **C)** A schematic of the reporters that express eGFP with native stop codon and premature stop codons (PTC) at the positions 60 and 120 (in scale). **D)** Flow cytometry analysis shows loss of eGFP fluorescence in the presence of a PTC. **E)** Northern blot of RNA samples of three clones expressing the different eGFP reporters. The membrane was probed for eGFP mRNAs and rRNA (loading control). The bar graph shows average quantified GFP mRNA levels relative to rRNA from the three independent clones. **F)** Relative RNA levels of the reporters upon UPF1 depletion using the auxin inducible degron (2 hours induction). The bar graph shows average values from triplicate experiments and corresponding error bars. *eGFP* RNA levels were quantified using RT-qPCR. Levels were normalised with GAPDH levels and calculated with the 2-ΔΔCT method. The average ΔCT values of the reporter with the native stop codon, without 5-Ph-IAA treatment, was used as the parental ΔCT values to calculate the 2-ΔΔCT values of all samples. **G)** The presence (coloured) or absence (not-coloured) of UPF1, 2 and 3 is shown schematically for Euglenozoa. The tree is based on 18S rDNA phylogeny data taken from [84]; the *Trypanosomatida* branch was manually added [85] and is not to scale.

earlier studies reporting instability of PTC-containing reporters [47,48,86]. Employing our AID2 degron system we could now test whether UPF1 depletion effects the stability of these PTC reporters. For technical reasons we had to switch from Northern blots to qPCR, but we observed a similar PTC-related decrease in mRNA levels as with northern blot, albeit slightly less severe for the early PTC reporter (by only 74% instead of 94%). To our surprise, depletion of UPF1 almost doubled the levels of the early PTC reporter, reaching 59% of the levels of the native GFP mRNA. This strongly suggests that UPF1-dependent NMD is functional (Fig 4F). Unexpectedly, the AID2 degron depletion of UPF1 also led to a mild but significant decrease of native eGFP mRNA levels by approximately 14% which could be a rather unspecific consequence triggered by the observed loss of fitness. This effect may have prevented that mRNA levels of the early-PTC reporter were fully restored to the levels of the native GFP. Moreover, it could also explain why late PTC mRNA levels remained unaffected by UPF1 depletion (Fig 4F). Another potential factor obscuring the rescue is the residual abundance of transcripts formed before complete UPF1 depletion during the 2h induction time, chosen to restrict secondary effects. Nevertheless, our data show that UPF1 contributes to NMD, but they do not fully rule out the possibility that additional factors contribute. Notably, UPF1 deletion in other organisms with an EJC-independent NMD, such as *S. pombe*, also does not completely restore the steady state levels of early PTC reporters [87]. We thus conclude that there is an EJC-independent NMD-like pathway in *T. brucei* which significantly relies on UPF1.

Homologues to UPF1, UPF2 and UPF3 are present across the eukaryotic kingdom (S9 Fig) and must have been present in the last eukaryotic common ancestor. Losses have since occurred in all branches of the eukaryotes with rather patchy distribution, mainly of UPF2 and UPF3, and very occasionally of the more conserved UPF1. Most of the few UPF1 losses are isolated, e.g., in single organisms within UPF1-encoding clades; these may be very recent losses or could be attributed to incomplete genome sequence information. The one notable exception is the group of Kinetoplastea: UPF3 is absent in all representative organisms, some have additionally lost UPF2 and others have lost all three UPF variants. Notably, the losses of UPF1 and UPF2 must have occurred independently on several occasions, for example in *T. congolense* and in the ancestor of the *Leishmaniinae* or in *Paratrypanosoma* (Fig 4G). The loss of UPF isoforms may be the direct consequence of the almost complete loss of introns that is characteristic for Trypanosomatids. Consistent with this hypothesis, all organisms of the two intron-rich sister groups of the Kinetoplastids, Diplonemids and Euglenids, possess all three UPF isoforms. Notably, unlike *cis*-splicing, *trans*-splicing does not create PTCs. In the (near) absence of the former, NMD may have become redundant, hence lifting the selective pressure to maintain the UPF1/2/3 genes.

In conclusion, we provide evidence for the presence of an EJC independent NMD-like pathway in *T. brucei* that is fully or partially dependent on UPF1. Based on the missing essentiality of UPF1 together with the occurrence of UPF1 losses within Kinetoplastida and the failure of UPF1 depletion to fully restore the stability of early PTC reporter RNAs (Fig 4) it remains possible that additional, UPF1-independent mechanisms exist, but this possibility requires further exploration. Ribosome processivity, for example, influences mRNA stability in trypanosomes [88] and yeast [89]. Given that abnormally long 3'-UTRs contribute to NMD triggering [90] a further open question how NMD is mediated in the presence of long 3'-UTRs present in many trypanosome genes [47].

## Conclusion

Magoh and Y14 are the least-conserved components of the core EJC (excluding the metazoan-specific BTZ). The majority of yeast strains have lost genes for Y14 and Magoh [77] and in the early branching *Tetrahymena* a Magoh knockout is viable [52]. Likewise, in trypanosomes, Magoh and Y14 lack essential amino acids for interaction with eIF4AIII [42] and our TurboID experiments showed no evidence for association of Magoh/Y14 with eIF4AIII or CWC22. A double knockout of Magoh and Y14 evoked only a minor reduction of fitness. While we cannot exclude essential EJC-unrelated functions for Magoh and Y14 in other life cycle stages not assessed here, the lack of association between Magoh/Y14 with eIF4AIII and CWC22 strongly argues against Magoh and Y14 being part of a canonical EJC.

In contrast, eIF4AIII is more conserved than Y14 and Magoh and, for example, present in all yeast strains which have lost Y14 and Magoh [77]. Both human eIF4AIII and the yeast orthologue Fal1 have conserved EJC-independent functions in 18S rRNA processing that depend on the direct interaction with the nucleolar protein NOM1/Sgd1p (human/yeast) [82,91]. In trypanosomes, the dominant nucleolar localization of eIF4AIII and its interaction with the trypanosome NOM1 homologue and several other rRNA-associated, nucleolar proteins strongly indicates that this function in ribosome biosynthesis is conserved. It remains unclear whether trypanosome eIF4AIII has additional functions. Its partial localization to the nucleoplasm and interaction with CWC22 indicates a potential role in splicing; however, it could also be a remnant of its former presence within an EJC that has become non-functional. While we show here that the dominant localization of *T. brucei* eIF4AIII is nuclear, we do observe a partial cytoplasmic localization of eIF4AIII, slightly increasing at starvation stress. Thus, it is possible that eIF4AIII is involved in RNA export, despite not being part of an EJC, and it remains to be explored whether this function is in mRNA or ribosome export and whether it is essential. In contrast to published immunoprecipitation studies [44], our TurboID experiments do not support interactions of the EJC proteins with the putative TREX complex component Sub2 or with the NTF2 domain-containing protein NTF2L. Overall, this contradiction remains unresolved, as both methods can produce artefacts as well as absences and we have previously observed major differences between TurboID and immunoprecipitation data, even when done in parallel in the same lab [61]. However, at least for NTF2L, the TurboID data may be more reliable, as it is consistent with the non-nuclear localization of the protein [80].

Taken together, our data argue strongly against the presence of a canonical EJC in trypanosomes. The lack of interaction between eIF4AIII and Y14/Magoh would prevent the inhibition of the eIF4AIII ATPase activity and consequently the stable association of eIF4AIII to RNA. Moreover, if Y14/Magoh do not bind to eIF4AIII; they cannot recruit the TREX-subunit Yra1 to speed up export or PYM to recruit ribosomes during translation. Indeed, orthologues of Yra1 and PYM are absent from trypanosome genomes. Thus, even though trypanosomes have conserved EJC components, the EJC complex does not assemble, and the proteins may be

considered vestigial products of evolution. The lack of selective pressure drives poor sequence conservation, and finally loss of protein interactions and loss of genes.

The absence of a canonical EJC, together with the loss of UPF3 indicates that any trypanosome NMD pathway must be EJC independent, like in *S. pombe* [87]. Many Kinetoplastea have lost UPF2 or both UPF2 and UPF1, offering some evidence for a loss of selective pressure to keep the NMD pathway, which is concomitant with the (almost) complete loss in introns. Consistent with this, UPF1-depleted cells have an almost normal phenotype, likely because PTC numbers are low in the absence of cis-splicing and the loss of the remaining Kinetoplastea UPF genes may simply be a question of time. Nevertheless, when PTCs are included in a reporter (or an ectopically expressed endogenous gene), mRNA abundance does decrease proportionally to the position on the ORF [47,48], even down to 5% [48]: this observation is difficult to account for without an NMD-like mechanism and, in fact, we were able to show the UPF1-dependency of this phenomenon, demonstrating the presence of an UPF1 dependent NMD-like pathway.

Evolutionary pressure has favored a compact and almost intron-less genome in the parasitic trypanosomatids. As a consequence, the canonical exon junction complex and the NMD pathway have become superfluous and are in the process of being lost. In *Trypanosoma brucei*, Y14 and Magoh homologues are still present but not part of an EJC and NMD is still functional, but not essential.

## Supporting information

**S1 Table. Plasmids, PCR products, oligos and FISH probes used in this work.**
(XLSX)

**S2 Table. Mass spectrometry data of the TurboID purifications (eIF4AIII, Magoh and Y14) and respective GO term enrichment analyses.**
(XLSX)

**S3 Table. Blast results and sequences of UPF1, UPF2 and UPF3 from a range of organisms representing the diversity of the eukaryotic lineages.**
(XLSX)

**S1A Fig. eIF4AIII sequence comparison.** Residues that are involved in the Magoh and Y14 interaction in the human protein (pdb 2XB2) are indicated by green and blue boxes below the alignment. Residues engaging in ATP (ANP (phosphoaminophosphonic acid-adenylate ester) in the crystal structure) binding have purple boxes. DEAD-box helicase motifs (Q-motif; DEAD box) are highlighted. Hs, *Homo sapiens* (P38919); Ce, *Caenorhabditis elegans* (Q9BL61); Dm, *Drosophila melanogaster* (Q9VHS8); Pf, *Plasmodium falciparum* (Q8IKF0); Sc, *Saccharomyces cerevisiae* (Q12099); Tb, *Trypanosoma brucei* (Tb927.11.8770); Tc, *Trypanosoma cruzi* (C4B63_6g161).
(PDF)

**S1B Fig. UPF1 sequence comparison.** Residues that are involved in UPF2 interaction in the human protein (PMID: 19556969) are indicated (green boxes below the alignment). Residues engaging in ATP binding have purple boxes. The DEAD-box helicase motif of UPF1 is highlighted. (Hs, *Homo sapiens* (Q92900); Ce, *Caenorhabditis elegans* (O76512); Dm, *Drosophila melanogaster* (Q9VYS3); Eg, *Euglena gracilis* (EG_transcript_1144); Sc, *Saccharomyces cerevisiae* (P30771); Tb, *Trypanosoma brucei* (Tb927.5.2140); Tc, *Trypanosoma cruzi* (C3747_24g302).
(PDF)

**S2 Fig. Y14, Magoh and Y14/Magoh knockouts: verification of the cell lines by PCR.** Cell cycle analysis.
(PDF)

**S3 Fig. Single molecule FISH for *DBP2B* (intron and exon) for Magoh/Y14 knockouts and UPF1 auxin inducible degron cells.**
(PDF)

**S4 Fig. Auxin inducible degron (eIF4AIII, NTF2L and UPF1): cell line verification by PCR.**
(PDF)

**S5 Fig. Cell cycle analysis of eIF4AIII auxin inducible degron cell lines.**
(PDF)

**S6A Fig. Representative images of eYFP fusions of Y14 after different treatments (untreated, PBS starvation, sinefungin, actinomycin D).**
(PDF)

**S6B Fig. Representative images of eYFP fusions of Magoh after different treatments (untreated, PBS starvation, sinefungin, actinomycin D).**
(PDF)

**S6C Fig. Representative images of eYFP fusions of eIF4AIII after different treatments (untreated, PBS starvation, sinefungin, actinomycin D).**
(PDF)

**S7 Fig. EJC proteins were fused to TurboID-HA tandem tag and imaged with fluorescent streptavidin and anti-HA (representative images with several cells).**
(PDF)

**S8 Fig. Auxin-inducible degron of NTF2L.**
(PDF)

**S9 Fig. Presence of UPF1, 2 and 3 in organisms representing the major branches of the eukaryotic lineages.**
(PDF)

**S1 Raw Images. Raw images for Fig 1B, Fig 4A, Fig 4E and S8 Fig.**
(PDF)

## Acknowledgments

We are grateful to the OMICS Proteomics BIOCEV core facility for excellent technical service and thank Jay Bangs for anti-BiP and to Silke Braune for excellent technical assistance.

## Author contributions

**Conceptualization:** Bernardo Papini Gabiatti, Susanne Kramer, Martin Zoltner.

**Data curation:** Susanne Kramer, Martin Zoltner.

**Formal analysis:** Bernardo Papini Gabiatti, Eden Ribeiro Freire, Fabiola Holetz, Mark Carrington, Susanne Kramer, Martin Zoltner.

**Funding acquisition:** Susanne Kramer, Martin Zoltner.

**Investigation:** Bernardo Papini Gabiatti, Eden Ribeiro Freire, Johanna Odenwald, Janaina de Freitas Nascimento, Fabiola Holetz, Mark Carrington, Susanne Kramer, Martin Zoltner.

**Methodology:** Bernardo Papini Gabiatti, Eden Ribeiro Freire, Johanna Odenwald, Janaina de Freitas Nascimento, Fabiola Holetz, Susanne Kramer, Martin Zoltner.

**Project administration:** Susanne Kramer, Martin Zoltner.

**Resources:** Mark Carrington.

**Validation:** Bernardo Papini Gabiatti, Susanne Kramer.

**Visualization:** Bernardo Papini Gabiatti, Eden Ribeiro Freire, Fabiola Holetz, Susanne Kramer, Martin Zoltner.

**Writing – original draft:** Bernardo Papini Gabiatti, Mark Carrington, Susanne Kramer, Martin Zoltner.

**Writing – review & editing:** Bernardo Papini Gabiatti, Eden Ribeiro Freire, Johanna Odenwald, Fabiola Holetz, Mark Carrington, Susanne Kramer, Martin Zoltner.

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
