## [Decision Letter · Decision Letter 0]

7 Jun 2024

PONE-D-24-18641Intron-loss in Kinetoplastea correlates with a non-functional EJC and loss of NMD factorsPLOS ONE

Dear Dr. Zoltner,

Thank you for submitting your manuscript to PLOS ONE. After careful consideration, we feel that it has merit but does not fully meet PLOS ONE’s publication criteria as it currently stands. Therefore, we invite you to submit a revised version of the manuscript that addresses the points raised during the review process.

As the comments below show, the three referees collectively agree that the manuscript presents well-controlled experiments with significant new results that are of interest to the field. Most of the comments from the referees can be addressed by re-writing parts of the manuscript. Please know that decision for major revision is simply to provide you opportunity to address the comment from referee #1 to use a reporter gene with a premature termination codon to test if UPF1 depletion down to 2% of the normal levels will inhibit NMD. Only a direct evidence from such an experiment can allow a clear conclusion that T brucei do not have an operational NMD pathway. In absence of such direct evidence, the conclusions need to be softened and the caveat acknowledged. Also, it will be important to address the comments from referee #2 regarding the discussion of previous experiments that arrived at different conclusions.

We look forward to receiving your revised manuscript.

Kind regards,

Guramrit Singh

Academic Editor

PLOS ONE

Journal Requirements:

   "The project was funded by a bilateral GACR/DFG grant (project IDs.: 21-19503J and KR4017/9-1; to M. Z. and S. K., respectively) and the DFG grant KR4017/8-1 to S.K. "

4. Please note that your Data Availability Statement is currently missing the repository name and/or the DOI/accession number of each dataset OR a direct link to access each database. If your manuscript is accepted for publication, you will be asked to provide these details on a very short timeline. We therefore suggest that you provide this information now, though we will not hold up the peer review process if you are unable.

Reviewers' comments:

Reviewer's Responses to Questions

**Comments to the Author**

1. Is the manuscript technically sound, and do the data support the conclusions?

Reviewer #1: Yes

Reviewer #2: Partly

Reviewer #3: Yes

2. Has the statistical analysis been performed appropriately and rigorously? 

Reviewer #1: Yes

Reviewer #2: Yes

Reviewer #3: Yes

3. Have the authors made all data underlying the findings in their manuscript fully available?

Reviewer #1: Yes

Reviewer #2: Yes

Reviewer #3: Yes

4. Is the manuscript presented in an intelligible fashion and written in standard English?

Reviewer #1: Yes

Reviewer #2: Yes

Reviewer #3: Yes

5. Review Comments to the Author

Reviewer #1: The study entitled “Intron-loss in Kinetoplastea correlates with a non-functional EJC and loss of NMD factors” by Gabiatti et al. primarily re-evaluates the presence, composition and functionality of potential exon junction complexes (EJCs) and the putative nonsense-mediated mRNA decay (NMD) machinery in Trypanosomes. The EJC plays several crucial roles in shaping and regulating the transcriptome of many metazoan organisms. Especially organisms with complex (alternative) splicing patterns rely on the EJC to mark successfully cis-spliced transcripts. EJCs assembled on RNAs influence a wide range of processes such as regulating alternative splicing, facilitating mRNA export, and helping to identify problematic transcripts during translation by triggering NMD. Due to the rather trypanosome-unique biological features concerning gene expression in these unicellular eukaryotic organisms, the existence and functional “conservation” of EJCs in trypanosomes is currently unclear. Although homologues of core EJC factors were found e.g. in Trypanosoma brucei, the necessity to assemble and deposit EJCs on cis-spliced transcripts is disputable since only two intron-containing genes are present in the genome.

In this study, the authors used conventional knockouts for non-essential EJC(-related) genes, as well as a state-of-the-art conditional degron system (AID2) to rapidly deplete other factors such as the EJC core protein eIF4AIII and the putative NMD core factor UPF1 in T. brucei. Using multiple experimental approaches including sub-cellular localization, growth analysis and proximity labeling-based proteomic interaction studies, the authors provide multiple evidence contradicting the presence and functionality of EJC in trypanosomes. Due to the essentiality, distinct localization, and interaction pattern with nucleolar proteins, eIF4AIII seems to be rather important for EJC-independent functions, potentially involving ribosome biosynthesis rather than splicing regulation. Additionally, the depletion of the UPF1 homologue in trypanosomes did neither influence cell growth, nor the levels of the potentially PTC-containing intron-retained DBP2B transcript, suggesting that NMD is not functional in T. brucei.

Overall, the study by Gabiatti et al. is of high quality, technically sound and further clarifies the (non )existence/function of EJCs and NMD machinery in trypanosomes. This contribution is important in my view, as it helps to assess the broader evolutionary scope of EJCs and NMD in eukaryotes, which is often centered on humans or mice. However, from a rather “mammalian-centric” perspective I have a few minor comments/questions that could potentially make the study more accessible and relevant to also non-trypanosome experts. Also, I have two rather major points concerning the experimental approach on UPF1 and NMD.

Major points:

• It is very understandable and technically elegant to use the AID2-degron approach to study the depletion of the essential eIF4AIII in trypanosomes. I was wondering why the degron approach was also used for UPF1, especially since it does not seem to be essential for cell growth. In other words, was a knockout of UPF1 attempted?

• Another potential issue with the characterization of UPF1 depletion is the obvious lack of endogenous PTC-containing (= potentially NMD-activating) mRNA targets. The authors have chosen to assess the DBP2B splicing pattern by single-molecule FISH, which showed no apparent difference in UPF1-depleted cells. Having another NMD target/reporter or experimental approach to support this finding would probably strengthen the conclusion that there is indeed no functional NMD in trypanosomes. Attempting constructive feedback, I was wondering whether:

o a) reporter genes (e.g. GPI-PLC) with PTCs could be employed to re-evaluate the observations made in the Delhi et al. (2011) study, using the technically more advanced AID2-system for UPF1 depletion? If there is indeed no NMD in T. brucei, the depletion of UPF1 should not affect the reporter mRNA levels.

o b) does intron-retention of the other intron-containing gene in T. brucei also lead to the formation of a PTC? In other words, could one use the other gene as “NMD reporter” as well?

o c) are there potential experimental alternatives to the single-molecule FISH approach to further underline the unchanged RNA isoform levels (e.g. intron-retained vs spliced)? Are “classical” RNA extraction, cDNA synthesis and PCR-based approaches potentially applicable here?

Minor points:

• Since I am not very familiar with Trypanosome-specific nomenclature, I was wondering how much the names of potential EJC components should/could be harmonized with the human nomenclature. Obviously, a general “renaming” is not an effort the study at hand has to necessarily provide, but it could help to make it clearer which gene/protein is referred to. For example, the HGNC approved symbols for BTZ is CASC3; for eIF4AIII is EIF4A3 and for Y14 is RBM8A (see https://www.genenames.org/data/genegroup/#!/group/1238).

• Sometimes the description and technical nomenclature could be more precise in my view, for example using “AID” or “AID2” consistently (since they are not the same system). The figures itself and the manuscript text often refers to “AID” although the AID2 system with the F74G mutation of OsTIR1 and 5-Ph-IAA instead of conventional IAA was used. The same applies to BioID (e.g. line 455) or TurboID.

• The description of BTZ (CASC3) as EJC core factor is debatable (see e.g. lines 21-23). Although many reviews still list it as “core” factor, evidence obtained from human cells and different labs indicate that CASC3 is rather a peripheral/cytoplasmic/auxiliary EJC factor (PMID: 32621609 & PMID: 30466796).

• The material and methods section contains many sentences simply referring to “standard procedures” (e.g. line 168) or “as described” (e.g. line 206-207). To make the methods section more understandable to a broader audience (i.e. non-trypanosome experts), I would recommend explaining those methods at least briefly.

• Concerning the growth plots (e.g. Fig 1A and C), I was wondering whether the underlying data was continuously monitored, or specific time-points were measured. In the latter case, I would include the individual data points to help estimate the spread.

• For Fig 1C: Could the PDT not be determined for these conditions?

• The authors referred to the fact that “both Magoh and Y14 lack amino acids required for eIF4AIII interaction” (lines 113-114). Along the same lines, I was wondering how specific catalytic/functional residues (e.g. of the helicase domain) in eIF4AIII and UPF1 are conserved between species (including for example yeast and humans)? If UPF1 is indeed redundant, has it retained or already lost key functional residues?

Reviewer #2: The aim of this study is to evaluate the role of T. brucei proteins containing domains conserved in other eukaryotic proteins that are described as core of EJC, a complex involved in cis-splicing and quality control pathways. The work focused on evaluating the interaction among these proteins as components of the same complex. As demonstrated by the authors, the proteins are not components of the same complex in T. brucei. In parallel, they observed that the core proteins of EJC (Magoh/Y14) and a conserved protein of the EJC-independent quality control pathway (UPF1) are not involved with processing of mRNA containing introns in this parasite. The work was well done, the figures are clear, the experiments well performed. Some results are not novelty since they were previously demonstrated by other groups, such as eIF4AIII, which is essential to the parasitic survival, and Magoh, Y14, NTF2L and UPF1 which are not. However, the authors also obtain contrasting results, different to what has been published before. They should be more specific in highlighting the novelty. In some cases, they try to explain the differences observed in previous studies by contesting or invalidating the data of other groups, that they are artifacts of technique. We are aware that artifacts can occur, but in my point of view, the way the authors lead the discussion is not appropriate. Instead of trying to discuss limitations and possible artifacts, or even invalidating results from other groups, they should consider the fact that there are significant differences in gene expression between trypanosome species. Therefore, not everything that is seen in T. brucei occurs in other species of the genus. For example, RNAi exists in T.brucei but does not exist in T.cruzi. T. brucei has protein-coding genes that are transcribed by RNA pol. I, whereas in T.cruzi, this is not seen. In addition, the infective forms act very differently, so they are biologically divergent, and this may reflect differences in the control of gene expression. It is not a rule to state that what is being seen in T. brucei should be generalized to all species of Trypanosoma. They could try to enrich the discussions of the results by trying to infer possible differences in the gene expression of these parasites, at least between T. brucei and T. cruzi, instead of trying to invalidate studies performed by other researchers. The discussion of some results is also very speculative. Below, there are some comments regarding necessary revisions to improve the manuscript conclusions and develop more appropriate and realistic contributions to the field.

Abstract:

It is a bit confusing regarding the main conclusions of the work. They cite that

"The presence of the three core EJC proteins and a potential BTZ homologue (Rbp25) in trypanosomes have been suggested as an adaptation of the EJC function to mark trans-spliced mRNAs." However, they didn’t test this hypothesis in the present work. Instead, they test the role of this proteins using a cis-spliced mRNA. The main conclusion should be described based mainly on confirming the absence of a canonical EJC or NMD pathway in cis-spliced mRNAs.

Lines 83-84, they should specify the nomenclature: yeast? Mammalian?

Line 91 to 96, "In Metazoan" is repetitive.

Line 133 to 148. It is a bit confusing, I could not understand the point, and the gap of knowledge is not clear. Suggest reviewing it.

Line 150: The justification or the main goal is not clear. They attest: "Here, we set out to reinvestigate the trypanosome EJC and NMD components, employing a range novel tools." Why did you decide to reinvestigate using novel tools? What kind of novelty would this bring to the area? I think the main novelty is analyzing the involvement of these proteins in cis-splicing. The conclusions addressing their role in trans-splicing are very speculative.

Line 153-154. The authors didn’t investigate the function of proteins using localization of trans-spliced mRNAs. They only obtained results using cis-spliced mRNAs (DBP2B). So, they should not attest: "Our data strongly indicate that the EJC has no function in the nuclear export of trans-spliced mRNAs."

It seems the authors based this conclusion on viability of mutant parasites and proteins interactions, but there is not enough evidence to affirm this.

Line 191: They should include the reference 49 in addition to 48, since ref. 48 is related only to NLS and it is the first time in the text that TurboID is mentioned.

"A cell line used as a negative control of the TurboID experiment expressed a codon-optimized eYFP-TurboID-HA with the La protein NLS (48) integrated into the tubulin locus."

Line 318: In the phrase "However, we noticed two differences between Y14/Magoh and eIF4AIII" they mean difference between the localization of these proteins? Or differences in relation to other eukaryotes proteins? It would clearer if this is specified.

Line 336: "For eIF4AIII (alias Hel46), our data contradict a range of immunofluorescence experiments done in T. cruzi and T. brucei, with antibodies against the native protein or anti-Protein A to a PTP-tagged version (65)." In this case, the authors should take into considerations that different analysis indicate the cytoplasmic localization of TceIF4AIII (hel45) in T. cruzi, such as fractions from sucrose density using only cytoplasmic fraction (hypotonic lyses are not enriched for nucleus), WB of nuclear and cytoplasmic fractions, ultrastructural microscopy using anti-Hel45 and WB blot of PTP-TAG using both anti-hel45 and anti-PTP. The difference in the localization is observed, indeed, in the T.brucei only when using antibodies raised against hel45. Maybe it is not a specific detection in T.brucei, making it a wrong interpretation in this publication. Yet, It doesn’t mean the data in T.cruzi are all artifact. Taking together your results, and the evidence published by Dhalia 2006 and Billington 2023, TbeIF4AIII is probably a nuclear protein. However, the authors should not invalidate all the evidence obtained in T.cruzi. It might reflect differences between the species. Besides, eIF4AIII is a shuttling protein in other eukaryotes and can have a global role of eIF4AIII in gene expression and independent of Y14 and MAGO, splicing in drosophila, for example (DOI: 10.7554/eLife.19881). So, It is hasty to exclude the role of eIF4AIII in cytoplasmic pathways in T.cruzi.

Line 376: "Moreover, the differences in localizations argue against the exclusive presence of the three proteins in one complex." Localization assays indicate but not confirm they are present in different complexes. The authors should affirm this evidence only in the section 3, protein interaction analysis.

Line 407 to 414. NTF2L is a protein which the function is still unknown in Trypanosomes, and it is not part of EJC in other eukaryotes. So, this discussion is out of the scope. The differences in the results are likely consequences of the versatilities of interaction assays and baits used in other works. Besides, both cited works didn’t affirm the presence of a canonic EJC in Tryps, they only demonstrated the association of proteins that contain conserved motives from EJC proteins, and it is not necessarily a direct interaction as demonstrated by the BioID approach. Besides, the non-essential function of NTF2L was already demonstrated elsewhere. Thus, the only novelty based on BioID is that NTF2L likely doesn’t interact directly to TbeIF4AIII, Magoh or Y14.

The section 4 is too speculative since the absence of a classical NMD pathway in Trypanosomes has already been proposed by another work https://doi.org/10.1371/journal.pone.0025112. In the previous work, the mRNA decay is not dependent of TbUPF1and TbUPf1 is not an essential protein in T.brucei. Thus, the novelty is that TbUPF1 is not also involved in cis-spliced decay.

Line 471-475 . The authors cannot prove that the cytoplasmic localization of TceIF4AIII is an artefact indeed. As explained above, there are several assays in T.cruzi to affirm the cytoplasmic localization in this parasite.

Line 481 - review this phrase. "At least for NTF2L, the BioID data may be more correct, as the protein is cytoplasmic (66) and the observed interaction with the EJC could be an artefact of the pulldown". There is still no evidence in the literature for a classical EJC, or that NTF2L is core component of EJC in Tryps. Besides NTF is a domain of Nuclear Transport Factor. Thus, the authors cannot exclude that NTF2L could be a shuttling protein and maybe be interacting dynamically with nuclear complexes.

Reviewer #3: This is a really nice study, which shows that the EJC has no apparent function in trypanosomes and indeed probably does not, as such, exist. I have only minor comments, just editing.

Generally: "that" defines. "which" describes.

The river that flows through Prague is called Vltava. ("that flows through Prague" is essential for the sentence to mean anything)

The Vltava river, which flows through Prague, was frozen. (The sentence is OK without "which flows through Prague".)

In this paper the authors use "that" way too much. ""that" should be replaced by "which" in lines 67, 69, 73, 88, 100, 295, 301, 463, 474, 513

Line 53: Impossibly vague. a) What is a "lineage" and (b) which other eukaryotes?

Line 56: What is a "regulatory region"? Regulating what?

line 60 "cell surface proteins that are transcribed by RNA polymerase I". Re-word.

line 61: There are no "long polycistronic mRNAs" because mRNA processing is co-transcriptional and quite fast. It's the transcription units that are polycistronic.

line 64: "polyadenylation of the upstream gene". Genes are not polyadenylated.

line 69 - which, not that.

line 72 - I don't think Mex67/Mtr2 can be described as a "transporter".

lline 89 - "this way" should be "thus"

line 90 - "the respective" should be "that RRM")

line 103 - Is it really? I suggest moving the discussion in lines 491-500 to the Introduction since they (and the references) are needed there.

line 151 - should be "range of novel tools". Also - faster than what?

line 153 - Use of BioID and antibody-independent protein localization are not novel.

line 157 - lack of a growth defect upon depletion of UPF1 does not conclusively demonstrate that it is not essential, only that a small amount suffices for normal growth. I suggest including "to 2% of normal".

line 293 - should be "could readily be deleted".

line 301 - "2 hours that was not reversed"

line 316 - confusing. What are "all three proteins", since the protein last described was DHH1?

line 326 - delete comma.

line 338 - replace "the authors report" by "the previous authors reported". Also rest of sentence in past tense.

lines 340, 343, 355 - delete comma.

line 345 delete comma after "is".

line 379 "associate" not "associates"

line 390 delete comma after "Magoh"

line 395 - should be "EJC-independently"

line 396 - typo (for for)

line 443 - delete comma.

line 461 1 - actually add commas: "and, for example, present"

line 467 - delete comma

line 476 - "its" should be either "it is" or "it's". ("its" indicates possession)

line 482 - Something is either correct or incorrect. It can't be "more correct". I suggest "more reliable".

lne 489 - "from trypanosome genomes" or "from trypanosomes' genomes".

line 503 - what is a "lower" eukaryote? (Evolution doesn't go from "low" to "high".) Please use a different term. Perhaps just "in all organisms apart from animals" (unless it's also found in plants).

line 516 - PTC not PCT; statement needs a reference. (Is this true in trypanosomes?)

line 521 - delete "a" at beginning.

Supplementary table S2 - some columns are completely empty (e.g. K, L, O, P). Suggest removal.

6. PLOS authors have the option to publish the peer review history of their article (what does this mean? ). If published, this will include your full peer review and any attached files.

**Do you want your identity to be public for this peer review?** For information about this choice, including consent withdrawal, please see our Privacy Policy .

Reviewer #1: **Yes: ** Volker Boehm

Reviewer #2: No

Reviewer #3: No

---

## [Author Response · Author response to Decision Letter 1]

7 Nov 2024

As the comments below show, the three referees collectively agree that the manuscript presents well-controlled experiments with significant new results that are of interest to the field. Most of the comments from the referees can be addressed by re-writing parts of the manuscript. Please know that decision for major revision is simply to provide you opportunity to address the comment from referee #1 to use a reporter gene with a premature termination codon to test if UPF1 depletion down to 2% of the normal levels will inhibit NMD. Only a direct evidence from such an experiment can allow a clear conclusion that T brucei do not have an operational NMD pathway. In absence of such direct evidence, the conclusions need to be softened and the caveat acknowledged. Also, it will be important to address the comments from referee #2 regarding the discussion of previous experiments that arrived at different conclusions.

We are highly grateful to R1 for his truly ‘constructive feedback’ which prompted us to test our AID2 degron UPF1 depletion for effects on PTC reporters. We have introduced an eGFP reporter gene with either an early PTC (position 60), a late PTC (position 120), or the complete ORF and can now conclude that trypanosomes have indeed an UPF1 dependent NMD pathway. Please find the detailed results below, in our response to R1 (page 4 of the rebuttal). These new data significantly improve our manuscript and turned our previous conclusion on the NMD pathway on the head. Consequently, we changed the title to:

Trypanosomes lack a canonical EJC but possess an UPF1 dependent NMD pathway

We have also addressed all other requests by the referees. Although we do not fully agree with all comments from R2 (see below), R2 overall has some valid points that we addressed by rewriting certain passages and softening our conclusions.

5. Review Comments to the Author

Reviewer #1: The study entitled “Intron-loss in Kinetoplastea correlates with a non-functional EJC and loss of NMD factors” by Gabiatti et al. primarily re-evaluates the presence, composition and functionality of potential exon junction complexes (EJCs) and the putative nonsense-mediated mRNA decay (NMD) machinery in Trypanosomes. The EJC plays several crucial roles in shaping and regulating the transcriptome of many metazoan organisms. Especially organisms with complex (alternative) splicing patterns rely on the EJC to mark successfully cis-spliced transcripts. EJCs assembled on RNAs influence a wide range of processes such as regulating alternative splicing, facilitating mRNA export, and helping to identify problematic transcripts during translation by triggering NMD. Due to the rather trypanosome-unique biological features concerning gene expression in these unicellular eukaryotic organisms, the existence and functional “conservation” of EJCs in trypanosomes is currently unclear. Although homologues of core EJC factors were found e.g. in Trypanosoma brucei, the necessity to assemble and deposit EJCs on cis-spliced transcripts is disputable since only two intron-containing genes are present in the genome.

In this study, the authors used conventional knockouts for non-essential EJC(-related) genes, as well as a state-of-the-art conditional degron system (AID2) to rapidly deplete other factors such as the EJC core protein eIF4AIII and the putative NMD core factor UPF1 in T. brucei. Using multiple experimental approaches including sub-cellular localization, growth analysis and proximity labeling-based proteomic interaction studies, the authors provide multiple evidence contradicting the presence and functionality of EJC in trypanosomes. Due to the essentiality, distinct localization, and interaction pattern with nucleolar proteins, eIF4AIII seems to be rather important for EJC-independent functions, potentially involving ribosome biosynthesis rather than splicing regulation. Additionally, the depletion of the UPF1 homologue in trypanosomes did neither influence cell growth, nor the levels of the potentially PTC-containing intron-retained DBP2B transcript, suggesting that NMD is not functional in T. brucei.

Overall, the study by Gabiatti et al. is of high quality, technically sound and further clarifies the (non )existence/function of EJCs and NMD machinery in trypanosomes. This contribution is important in my view, as it helps to assess the broader evolutionary scope of EJCs and NMD in eukaryotes, which is often centered on humans or mice. However, from a rather “mammalian-centric” perspective I have a few minor comments/questions that could potentially make the study more accessible and relevant to also non-trypanosome experts. Also, I have two rather major points concerning the experimental approach on UPF1 and NMD.

Major points:

• It is very understandable and technically elegant to use the AID2-degron approach to study the depletion of the essential eIF4AIII in trypanosomes. I was wondering why the degron approach was also used for UPF1, especially since it does not seem to be essential for cell growth. In other words, was a knockout of UPF1 attempted?

We considered, but in the end, we didn’t do it, as we thought, this wouldn’t add much, but yes, this would have been the ultimate proof for non-function.

• Another potential issue with the characterization of UPF1 depletion is the obvious lack of endogenous PTC-containing (= potentially NMD-activating) mRNA targets. The authors have chosen to assess the DBP2B splicing pattern by single-molecule FISH, which showed no apparent difference in UPF1-depleted cells. Having another NMD target/reporter or experimental approach to support this finding would probably strengthen the conclusion that there is indeed no functional NMD in trypanosomes. Attempting constructive feedback, I was wondering whether:

We had indeed done the FISH with both cis-spliced mRNAs, with no major difference, and since these were negative data, and the other RNA had some issues with low expression level, we only included DBP2B.

o a) reporter genes (e.g. GPI-PLC) with PTCs could be employed to re-evaluate the observations made in the Delhi et al. (2011) study, using the technically more advanced AID2-system for UPF1 depletion? If there is indeed no NMD in T. brucei, the depletion of UPF1 should not affect the reporter mRNA levels.

o b) does intron-retention of the other intron-containing gene in T. brucei also lead to the formation of a PTC? In other words, could one use the other gene as “NMD reporter” as well?

o c) are there potential experimental alternatives to the single-molecule FISH approach to further underline the unchanged RNA isoform levels (e.g. intron-retained vs spliced)? Are “classical” RNA extraction, cDNA synthesis and PCR-based approaches potentially applicable here?

We would like to thank R1 for this truly ‘constructive feedback’ which prompted us to test our AID2 degron UPF1 depletion for effects on PTC reporters. We have introduced an eGFP reporter gene with either an early PTC (position 60), a late PTC (position 120), or the complete ORF and can now conclude that trypanosomes have indeed an UPF1 dependent NMD pathway. Below is the description of these experiments. Naturally, we have changed other parts of the manuscript and the title as these new results admittedly turned our previous conclusion on the NMD pathway on the head.

Next, we expressed an eGFP reporter gene with either an early PTC (position 60), a late PTC (position 120), or the complete ORF (native, stop at position 240) constitutively from the tubulin locus (Fig. 4C). While both premature stop codons resulted in the expected loss of GFP protein (Fig. 4D), northern blotting revealed that the presence of a PTC significantly reduced the reporter mRNA levels in a position specific manner (Fig. 4E): Early and late PTC evoked approximate mRNA decreases of 94% and 37%, respectively, when compared to the mRNA encoding native eGFP. This is consistent with earlier studies reporting instability of PTC-containing reporters (46, 47, 84). Employing our AID2 degron system we could now test whether UPF1 depletion effects the stability of these PTC reporters. For technical reasons we had to switch from Northern blots to qPCR, but we observed a similar PTC-related decrease in mRNA levels as with northern blot, albeit slightly less severe for the early PTC reporter (by only 74% instead of 94%). To our surprise, depletion of UPF1 almost doubled the levels of the early PTC reporter, reaching 59% of the levels of the native GFP mRNA. This strongly suggests that UPF1-dependent NMD is functional (Fig. 4F). Unexpectedly, the AID2 degron depletion of UPF1 also led to a mild but significant decrease of native eGFP mRNA levels by approximately 14% which could be a rather unspecific consequence triggered by the observed loss of fitness. This effect may have prevented that mRNA levels of the early-PTC reporter were fully restored to the levels of the native GFP. Moreover, it could also explain why late PTC mRNA levels remained unaffected by UPF1 depletion (Figure 4F). Our data show that UPF1 contributes to NMD, but they do not fully rule out the possibility that additional factors contribute. Notably, UPF1 deletion in other organisms with an EJC-independent NMD, such as S. pombe, also does not completely restore the steady state levels of early PTC reporters (85). We thus conclude that there is an EJC-independent NMD pathway in T. brucei which significantly relies on UPF1.

Minor points:

• Since I am not very familiar with Trypanosome-specific nomenclature, I was wondering how much the names of potential EJC components should/could be harmonized with the human nomenclature. Obviously, a general “renaming” is not an effort the study at hand has to necessarily provide, but it could help to make it clearer which gene/protein is referred to. For example, the HGNC approved symbols for BTZ is CASC3; for eIF4AIII is EIF4A3 and for Y14 is RBM8A (see https://www.genenames.org/data/genegroup/#!/group/1238).

It is the usual problem with protein/gene nomenclature but using the mammalian protein names would possibly lead to more confusion. There is a defined nomenclature for trypanosome gene names (Clayton, C. E. et al. Genetic nomenclature for Trypanosoma and Leishmania. 97, 221–224 (1998).) that we are using and that is consistent with the trypanosome genome database. We had defined the mammalian nomenclature on the first call-out of EJC components in the introduction and have now added EIF4A3.

The EJC consists of three core subunits: the DExH/D-box RNA helicase eIF4AIII (EIF4A3/Fal1/ DDX48), Y14 (RBM8A) and Magoh; a fourth subunit BTZ (Barentsz/MNL51/CASC3), suggested as peripheral EJC factor (21, 22)is metazoan specific.

• The description of BTZ (CASC3) as EJC core factor is debatable (see e.g. lines 21-23). Although many reviews still list it as “core” factor, evidence obtained from human cells and different labs indicate that CASC3 is rather a peripheral/cytoplasmic/auxiliary EJC factor (PMID: 32621609 & PMID: 30466796).

We have corrected the description of BTZ as peripheral EJC factor- see above.

• Sometimes the description and technical nomenclature could be more precise in my view, for example using “AID” or “AID2” consistently (since they are not the same system). The figures itself and the manuscript text often refers to “AID” although the AID2 system with the F74G mutation of OsTIR1 and 5-Ph-IAA instead of conventional IAA was used. The same applies to BioID (e.g. line 455) or TurboID.

Thanks – we are now using TurboID and AID2 consistently

• The material and methods section contains many sentences simply referring to “standard procedures” (e.g. line 168) or “as described” (e.g. line 206-207). To make the methods section more understandable to a broader audience (i.e. non-trypanosome experts), I would recommend explaining those methods at least briefly.

We’ve added a brief description of the immunofluorescence/streptavidin stain method and added some detail about trypanosome transfection.

• Concerning the growth plots (e.g. Fig 1A and C), I was wondering whether the underlying data was continuously monitored, or specific time-points were measured. In the latter case, I would include the individual data points to help estimate the spread.

The growth was not measured continuously, and we have now added the data points.

• For Fig 1C: Could the PDT not be determined for these conditions?

The growth curve in Figure 1A is from knock-out cells, these have a constant growth rate. In Figure 1C, the protein depletion is induced by auxin and the growth rate decreases over time, therefore, a PDT is not meaningful.

• The authors referred to the fact that “both Magoh and Y14 lack amino acids required for eIF4AIII interaction” (lines 113-114). Along the same lines, I was wondering how specific catalytic/functional residues (e.g. of the helicase domain) in eIF4AIII and UPF1 are conserved between species (including for example yeast and humans)? If UPF1 is indeed redundant, has it retained or already lost key functional residues?

We have added Figure S1 with a multiple sequence alignment of eIF4AIII and UPF1, and functional residues as well as interacting residues highlighted.

Reviewer #2: The aim of this study is to evaluate the role of T. brucei proteins containing domains conserved in other eukaryotic proteins that are described as core of EJC, a complex involved in cis-splicing and quality control pathways. The work focused on evaluating the interaction among these proteins as components of the same complex. As demonstrated by the authors, the proteins are not components of the same complex in T. brucei. In parallel, they observed that the core proteins of EJC (Magoh/Y14) and a conserved protein of the EJC-independent quality control pathway (UPF1) are not involved with processing of mRNA containing introns in this parasite. The work was well done, the figures are clear, the experiments well performed. Some results are not novelty since they were previously demonstrated by other groups, such as eIF4AIII, which is essential to the parasitic survival, and Magoh, Y14, NTF2L and UPF1 which are not. However, the authors also obtain contrasting results, different to what has been published before. They should be more specific in highlighting the novelty. In some cases, they try to explain the differences observed in previous studies by contesting or invalidating the data of other groups, that they are artifacts of technique. We are aware that artifacts can occur, but in my point of view, the way the authors lead the discussion is not appropriate. Instead of trying to discuss limitations and possible artifacts, or even invalidating results from other groups, they should consider the fact that there are significant differences in gene expression between trypanosome species. Therefore, not everything that is seen in T. brucei occurs in other species of the genus. For example, RNAi exists in T.brucei but does not exist in T.cruzi. T. brucei has protein-coding genes that are transcribed by RNA pol. I, whereas in T.cruzi, this is not seen. In addition, the infective forms act very differently, so they are biologically divergent, and this may reflect differences in the control of gene expression. It is not a rule to state that what is being seen in T. brucei should be generalized to all species of Trypanosoma. They could try to enrich the discussions of the results by trying to infer possible differences in the gene expression of these parasites, at least between T. brucei and T. cruzi, instead of trying to invalidate studies performed by other researchers. The discussion of some results is also very speculative. Below, there are some comments regarding necessary revisions to improve the manuscript conclusions and develop more appropriate and realistic contributions to the field.

Abstract:

It is a bit confusing regarding the main conclusions of the work. They cite that

"The presence of the three cor

---

## [Decision Letter · Decision Letter 1]

19 Nov 2024

PONE-D-24-18641R1Trypanosomes lack a canonical EJC but possess an UPF1 dependent NMD pathwayPLOS ONE

Dear Dr. Zoltner,

Thank you for submitting your manuscript to PLOS ONE. After careful consideration, we feel that it requires some minor edits to the text based on new comments from reviewers 1 and 3. Therefore, we invite you to submit a revised version of the manuscript that addresses the points raised during the review process.

The revised manuscript will not need to go out for another round of reviews and the final decision will be made at the editorial level. So, we expect a quick turnaround for decision on the next submission.

We look forward to receiving your revised manuscript.

Kind regards,

Guramrit Singh

Academic Editor

PLOS ONE

Journal Requirements:

Reviewers' comments:

Reviewer's Responses to Questions

**Comments to the Author**

1. If the authors have adequately addressed your comments raised in a previous round of review and you feel that this manuscript is now acceptable for publication, you may indicate that here to bypass the “Comments to the Author” section, enter your conflict of interest statement in the “Confidential to Editor” section, and submit your "Accept" recommendation.

Reviewer #1: All comments have been addressed

Reviewer #2: All comments have been addressed

Reviewer #3: (No Response)

2. Is the manuscript technically sound, and do the data support the conclusions?

Reviewer #1: Yes

Reviewer #2: Yes

Reviewer #3: Yes

3. Has the statistical analysis been performed appropriately and rigorously? 

Reviewer #1: Yes

Reviewer #2: Yes

Reviewer #3: Yes

4. Have the authors made all data underlying the findings in their manuscript fully available?

Reviewer #1: Yes

Reviewer #2: Yes

Reviewer #3: Yes

5. Is the manuscript presented in an intelligible fashion and written in standard English?

Reviewer #1: Yes

Reviewer #2: Yes

Reviewer #3: Yes

6. Review Comments to the Author

Reviewer #1: The authors have adequately addressed all previous comments from my side, either by including new data or by proper explanation. I have no further major comments and recommend publication of this manuscript.

Nevertheless, I would like to use the opportunity to comment on the added NMD reporter data:

I am particularly pleased that the new experiments using an eGFP-based NMD reporter were insightful and helped to clarify the "NMD capacity" of trypanosomes. It seems that trypanosomes are in principle able to degrade PTC-containing mRNAs in an UPF1-dependent manner. The clear distinction whether the degradation is due to "canonical NMD" or other UPF1- and translation-dependent processes is not trivial and would require more detailed investigation (not in this manuscript though). One could even argue how NMD would be defined under these circumstances (e.g. does it require UPF2? Is the helicase activity of UPF1 required? Etc.). In light of these open questions and since the NMD(-like) activity seems to be only relevant for reporter mRNAs, the authors may want to soften some strong statements about the presence of an NMD pathway (e.g. in the new title).

A technical detail I wanted to comment on:

If I understand it correctly, the UPF1 depletion was performed for 2 hours before harvesting and extracting the RNA. I am not familiar with transcription kinetics in trypanosomes, but I was simply wondering whether this timeframe was maybe too short to see the "full" rescue of the early (position 60) PTC reporter? I certainly do not ask for those experiments to be repeated, but it could be another point for discussing the apparently incomplete rescue.

Reviewer #2: The authors have adequately addressed the comments by the reviwers in the revised version of the manuscript. The manuscript is improved and included new results to base their conclusions. Therefore, have no further comments.

Reviewer #3: The authors have addressed my comments and also introduced interesting new observations which change some of the conclusions. They are however going to have to revise this manuscript because the Abstract is 373 words long but the guidelines restrict it to 300 words.

Because of some newly-introduced parts, there is also a small number of new issues to address:

If there really is UPF-dependent NMD the authors need to address, at least very briefly,

(a) how this is consistent with the results in ref 48 and

(b) how it might be triggered, given the presence of some very long 3'-UTRs in Kinetoplastid mRNAs and the fact that previous reporter experiments (ref 46) were difficult to interpret. (I realise that the answer to the latter might be "we have no idea".)

Minor text issues (just sound awkward)

The section about TREX is confusing because it is now internally contradictory. I suggest starting with "Trypanosomes lack obvious homologues of most proteins of the TREX..." instead of "Trypanosomes lack homologues of most proteins of the TREX..."

"Consistent, UPF1 depletion has no major phenotype.." needs to be changed - "Consistent with this, UPF1-depleted cells have an almost normal phenotype" ("Consistent" can't be used on its own, and ll cells have a phenotype.....)

"As a consequence, the presence of a canonical exon junction complex, as well as an NMD pathway have become redundant and are in the process of being lost." - I suggest instead "As a consequence, the canonical exon junction complex and the NMD pathway have become redundant and are in the process of being lost." ("Superfluous" might also be better than "redundantP>

7. PLOS authors have the option to publish the peer review history of their article (what does this mean? ). If published, this will include your full peer review and any attached files.

**Do you want your identity to be public for this peer review?** For information about this choice, including consent withdrawal, please see our Privacy Policy .

Reviewer #1: **Yes: ** Volker Boehm

Reviewer #2: No

Reviewer #3: No

---

## [Author Response · Author response to Decision Letter 2]

21 Nov 2024

6. Review Comments to the Author

Reviewer #1: The authors have adequately addressed all previous comments from my side, either by including new data or by proper explanation. I have no further major comments and recommend publication of this manuscript.

Nevertheless, I would like to use the opportunity to comment on the added NMD reporter data:

I am particularly pleased that the new experiments using an eGFP-based NMD reporter were insightful and helped to clarify the "NMD capacity" of trypanosomes. It seems that trypanosomes are in principle able to degrade PTC-containing mRNAs in an UPF1-dependent manner. The clear distinction whether the degradation is due to "canonical NMD" or other UPF1- and translation-dependent processes is not trivial and would require more detailed investigation (not in this manuscript though). One could even argue how NMD would be defined under these circumstances (e.g. does it require UPF2? Is the helicase activity of UPF1 required? Etc.). In light of these open questions and since the NMD(-like) activity seems to be only relevant for reporter mRNAs, the authors may want to soften some strong statements about the presence of an NMD pathway (e.g. in the new title).

We agree that there are several open questions remaining about the precise role of UPF1 in the degradation of PTC-containing mRNAs and thus, overall, the definition of the NMD pathway in trypanosomes (and likewise other unicellular organisms such as yeast). We therefore have softened the title by referring to an NMD-like pathway:

‘Trypanosomes lack a canonical EJC but possess an UPF1 dependent NMD-like pathway’

Same in the discussion:

Notably, UPF1 deletion in other organisms with an EJC-independent NMD, such as S. pombe, also does not completely restore the steady state levels of early PTC reporters (85). We thus conclude that there is an EJC-independent NMD-like pathway in T. brucei which significantly relies on UPF1.

In conclusion, we provide evidence for the presence of an EJC independent NMD-like pathway in T. brucei that is fully or partially dependent on UPF1.

Further, we have added more discussion of this matter (see below, response to R3) and compare our findings critically to previous observations.

A technical detail I wanted to comment on:

If I understand it correctly, the UPF1 depletion was performed for 2 hours before harvesting and extracting the RNA. I am not familiar with transcription kinetics in trypanosomes, but I was simply wondering whether this timeframe was maybe too short to see the "full" rescue of the early (position 60) PTC reporter? I certainly do not ask for those experiments to be repeated, but it could be another point for discussing the apparently incomplete rescue.

Yes correct – we chose 2 hours depletion to restrict secondary effects of the auxin depletion. This was working for the early PTC (with abundance almost doubling, although levels were not fully restored). Simultaneously, we were clearly observing a general effect on the PTC-independent transcript abundance (which we coin to the global loss of fitness upon depletion). Due to the latter effect, we refrained from testing longer depletion times. We agree that residual transcripts made before the onset of full UPF1 depletion (and not completely turned over) may be responsible for the incomplete rescue and added the following to the discussion:

Another potential factor obscuring the rescue is the residual abundance of transcripts formed before complete UPF1 depletion during the 2h induction time, chosen to restrict secondary effects. Nevertheless, our data show that UPF1 contributes to NMD, but they do not fully rule out the possibility that additional factors contribute.

Reviewer #2: The authors have adequately addressed the comments by the reviwers in the revised version of the manuscript. The manuscript is improved and included new results to base their conclusions. Therefore, have no further comments.

Reviewer #3: The authors have addressed my comments and also introduced interesting new observations which change some of the conclusions. They are however going to have to revise this manuscript because the Abstract is 373 words long but the guidelines restrict it to 300 words.

Thanks for spotting – we shortened the abstract.

Because of some newly-introduced parts, there is also a small number of new issues to address:

If there really is UPF-dependent NMD the authors need to address, at least very briefly,

(a) how this is consistent with the results in ref 48 and

We see the data interpretation in Ref 48 (PMID: 34541528; cited in the introduction) as oversimplified for the reasons listed below.

1. NMD is not that efficient, so there will be some mRNA, even in humans 25% avoids turnover

2. The effectiveness of NMD varies from one mRNA to another,

3. Where there is a frame shift (first example figure 6 in the paper) no account is taken of the consequent change in codon use

4. The allocation of reads cannot be that accurate, looking at the second example in Figure 6 the abundance (RPM) in regions 4 and 5 are reversed. This is difficult to explain if the measurements are accurate. It would have been better to sum the two sets of allele specific RNAseq reads

5. There are some examples in figures S9 and S10 where the PTC allele is less abundant.

6. Some of the mRNAs chosen are developmentally regulated, which may have an effect

However, we feel that we are not in position (and don’t intend) to attack these results as our data is restricted to a reporter gene. For this, we show that PTCs reduce transcript levels in a position specific manner (Northern blot; Fig 4E) and that the drop in early PTC transcripts can be partly rescued by UPF-1 depletion (Fig 4F).

(b) how it might be triggered, given the presence of some very long 3'-UTRs in Kinetoplastid mRNAs and the fact that previous reporter experiments (ref 46) were difficult to interpret. (I realise that the answer to the latter might be "we have no idea".)

Thanks, we have added a sentence to the discussion.

Given that abnormally long 3'-UTRs contribute to NMD triggering (88) a further open question how NMD is mediated in the presence of long 3'-UTRs present in many trypanosome genes (46).

Minor text issues (just sound awkward)

The section about TREX is confusing because it is now internally contradictory. I suggest starting with "Trypanosomes lack obvious homologues of most proteins of the TREX..." instead of "Trypanosomes lack homologues of most proteins of the TREX..."

Thanks. Done.

"Consistent, UPF1 depletion has no major phenotype.." needs to be changed - "Consistent with this, UPF1-depleted cells have an almost normal phenotype" ("Consistent" can't be used on its own, and ll cells have a phenotype.....)

Agree. Changed.

"As a consequence, the presence of a canonical exon junction complex, as well as an NMD pathway have become redundant and are in the process of being lost." - I suggest instead "As a consequence, the canonical exon junction complex and the NMD pathway have become redundant and are in the process of being lost." ("Superfluous" might also be better than "redundantP>

Thanks. Done.

---

## [Editor Report · Decision Letter 2]

29 Nov 2024

Trypanosomes lack a canonical EJC but possess an UPF1 dependent NMD-like pathway

PONE-D-24-18641R2

Dear Dr. Zoltner,

We’re pleased to inform you that your manuscript has been judged scientifically suitable for publication and will be formally accepted for publication once it meets all outstanding technical requirements.

Kind regards,

Guramrit Singh

Academic Editor

PLOS ONE

Additional Editor Comments (optional):

Please review the text for spellings and typos as a quick read identified these issues:

-EJC is typed "ECJ" in abstract (last sentence)

-“ophistokonts” and “opisthokonts” spellings used in Introduction

-Both nonsense and non-sense are used in text interchangably
---

## [Editor Report · Acceptance letter]

PONE-D-24-18641R2

PLOS ONE

Dear Dr. Zoltner,

I'm pleased to inform you that your manuscript has been deemed suitable for publication in PLOS ONE. Congratulations! Your manuscript is now being handed over to our production team.

Kind regards,

on behalf of

Dr. Guramrit Singh

Academic Editor

PLOS ONE